# Profiling of the plasma proteome across different stages of human heart failure

Anna Egerstedt[1], John Berntsson [1,2], Maya Landenhed Smith[3], Olof Gidlöf [1], Roland Nilsson [4], Mark Benson[5], Quinn S. Wells[6], Selvi Celik [1], Carl Lejonberg[1], Laurie Farrell[5], Sumita Sinha[5], Dongxiao Shen[5], Jakob Lundgren[1,7], Göran Rådegran[1,7], Debby Ngo[5], Gunnar Engström[2], Qiong Yang [8], Thomas J. Wang[6], Robert E. Gerszten[5,9] & J. Gustav Smith[1,7,9,10]*

Heart failure (HF) is a major public health problem characterized by inability of the heart to maintain sufficient output of blood. The systematic characterization of circulating proteins across different stages of HF may provide pathophysiological insights and identify therapeutic targets. Here we report application of aptamer-based proteomics to identify proteins associated with prospective HF incidence in a population-based cohort, implicating modulation of immunological, complement, coagulation, natriuretic and matrix remodeling pathways up to two decades prior to overt disease onset. We observe further divergence of these proteins from the general population in advanced HF, and regression after heart transplantation. By leveraging coronary sinus samples and transcriptomic tools, we describe likely cardiac and specific cellular origins for several of the proteins, including Nt-proBNP, thrombospondin-2, interleukin-18 receptor, gelsolin, and activated C5. Our findings provide a broad perspective on both cardiac and systemic factors associated with HF development.

[1] Department of Cardiology, Clinical Sciences, Lund University, Lund, Sweden. [2] Cardiovascular Epidemiology, Clinical Sciences, Lund University, Malmö, Sweden. [3] Department of Cardiothoracic Surgery, Clinical Sciences, Lund University and Skåne University Hospital, Lund, Sweden. [4] Department of Medicine, Karolinska Institutet and Karolinska University Hospital, Stockholm, Sweden. [5] Division of Cardiovascular Medicine, Beth Israel Deaconess Medical Center and Harvard Medical School, Boston, MA, USA. [6] Division of Cardiovascular Medicine, Vanderbilt University, Nashville, TN, USA. [7] Department of Heart Failure and Valvular Disease, Skåne University Hospital, Lund, Sweden. [8] Department of Biostatistics, Boston University School of Public Health, Boston, MA, USA. [9] Program in Medical and Population Genetics, Broad Institute of Harvard and Massachusetts Institute of Technology, Cambridge, MA, USA. [10] Wallenberg Center for Molecular Medicine and Lund University Diabetes Center, Lund University, Lund, Sweden. *email: gustav.smith@med.lu.se

Heart failure (HF) is the end-stage condition of all heart disease, characterized by the inability of the heart to maintain sufficient output of blood for the demands of the body at normal filling pressures. This condition is highly prevalent and confers increased mortality, risk of deadly arrhythmias, impaired quality of life, and frequent need for hospitalization because of breathing difficulties and fluid accumulation[1,2]. HF results from cardiac overload or injury in combination with activation of neurohormonal, metabolic, inflammatory and other pathways[3]. In particular, circulating levels of angiotensin II, aldosterone, catecholamines and natriuretic peptides are all increased in heart failure, associated with increased mortality [4], and constitute the principal targets for HF therapy[1]. These targets were originally identified from candidate-based physiological studies. Global plasma profiles of HF have previously not been established, but could facilitate identification of novel preventive and therapeutic targets[5].

The sequencing of the human genome and subsequent identification of protein-coding genes resulted in a catalog of approximately 20,000 proteins[6], of which a few thousand have been detected in plasma[7–9]. Recently, robust affinity-based methods to systematically assay established proteins and pathways have been developed, with sufficient sensitivity to detect proteins across the entire range from the most highly abundant protein albumin (present in human plasma in 35−45 mg/mL), which constitutes 55% of the plasma protein mass[10], through medium-abundance proteins such as C-reactive protein (CRP, present in a few ng/mL), to low-abundance proteins such as natriuretic peptides (present in a few pg/mL). Of affinity-based methods, an aptamer-based system has been demonstrated to be particularly robust for deep proteome coverage and high sample throughput to address large-scale human studies[10–12]. HF represents a particularly promising application of such a platform, given the key role of circulating proteins in the pathophysiology of this condition as described above, and the accessibility of plasma samples. Here, we therefore apply this aptamer-based assay for proteomic profiling of plasma from cohorts representing different stages of HF including early HF development, manifest advanced HF, and reversal of HF after heart transplantation. Our results provide a comprehensive perspective on the plasma proteome in HF and identify both cardiac and extracardiac circulating proteins associated with HF incidence which display a graded increase across nonfailing, early HF and advanced HF populations.

## Results

**Baseline characteristics of study cohorts.** Plasma samples from three cohorts underwent proteomic analysis (Fig. 1): (1) a nested case-cohort subset from the population-based Malmö Diet and Cancer Study cardiovascular cohort (MDC-CC) consisting of all cases of incident HF ($n = 185$) during a median follow-up of 20.2 years (range 0.36−22.2; median time to diagnosis 14.6 [interquartile range 9.9−17.8]) and a random population-representative subset from the same cohort ($n = 583$), (2) a hospital-based cohort of 84 patients with manifest HF, and (3) serial samples from 30 patients with advanced HF undergoing heart transplantation. Subjects with incident HF had higher body mass index, higher blood pressure, and a higher prevalence of diabetes, atrial fibrillation and a history of myocardial infarction as compared to subjects without incident HF. Subjects with incident HF also had higher plasma concentrations of N-terminal pro-B-type Natriuretic Peptide (Nt-proBNP), CRP and creatinine based on clinical assays. The risk estimates for incident HF of individual risk factors in the nested subcohort, based on Prentice-weighted Cox proportional hazards regression, were similar to unweighted estimates from the full MDC-CC (Supplementary Table 1). The hospital-based sample of manifest HF patients had a lower age at diagnosis than the population-based incident cases from the MDC-CC, and had a lower burden of comorbid conditions (Table 1). As expected, the Nt-proBNP concentration was >10-fold higher in manifest HF than at baseline in cases with incident HF from the MDC-CC. The advanced HF patients waiting for a heart transplant had further lower age, burden of comorbidities and higher Nt-proBNP. Additional clinical characteristics of the manifest HF patients and heart transplant recipients are presented in Supplementary Table 2.

**Protein characteristics and pathway representation.** The aptamer-based assay measures 1305 proteins, which are listed in

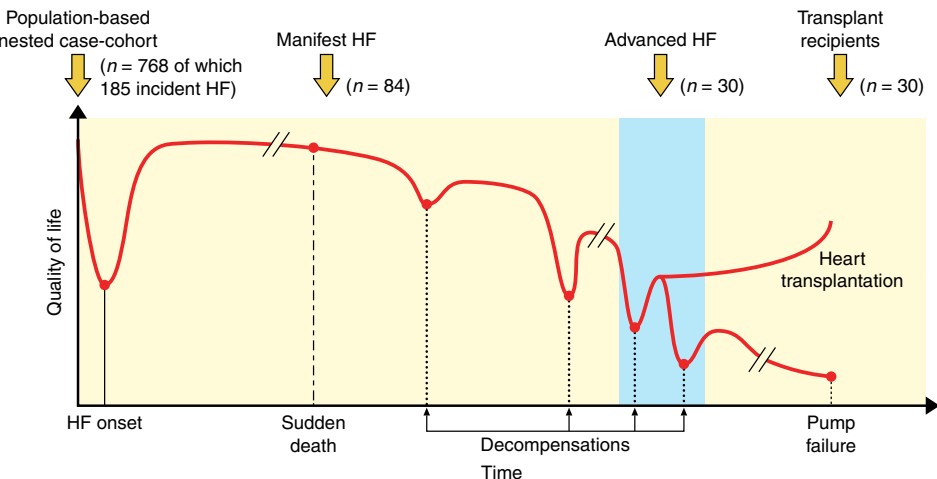

**Fig. 1 Sampling of study cohorts from different stages of heart failure.** Schematic representation of the natural history of heart failure (HF) progression, from the initial timepoint of disease onset through a gradual decline with increasing episodes of worsening typically necessitating in-hospital care (decompensations) towards terminal pump failure. Cohorts for the current study were obtained from four stages: before disease onset (population-based cohort), during manifest HF, and serially in advanced HF before and after heart transplantation. Proteomic profiles in subjects with incident HF from the population-based cohort were compared to subjects without incident HF. Manifest HF patients were compared to the population-based cohort. Samples from heart transplant recipients were compared to the corresponding samples from the same patient before transplantation.

**Table 1 Baseline characteristics.**

| | Full MDC-CC (n = 6103) | Nested MDC-CC subcohort (n = 583) | Incident MDC-CC HF cases (n = 185) | Manifest HF (n = 84) | Heart transplant recipients (n = 30) |
|---|---|---|---|---|---|
| Baseline age (years) | 57.48 (5.93) | 56.97 (5.76) | 61.25 (4.76) | 65.79 (12.04) | 51.14 (9.97) |
| Age at diagnosis (years) | 74.75 (6.93) | — | 74.71 (6.93) | — | — |
| Male sex (n [%]) | 2572 (42.14%) | 237 (40.58%) | 106 (56.99%) | 65 (77.38%) | 23 (76.7%) |
| BMI (kg/m$^2$) | 25.77 (3.99) | 25.42 (3.69) | 27.76 (4.49) | 27.66 (6.24) | 26.52 (4.40) |
| SBP (mm Hg) | 141.38 (19.08) | 138.53 (18.11) | 149.31 (19.90) | 123.85 (20.80) | 100.20 (13.56) |
| DBP (mm Hg) | 86.98 (9.46) | 85.52 (9.00) | 89.96 (9.47) | 74.90 (11.95) | 71.55 (9.96) |
| Antihypertensive treatment (n [%]) | 1115 (18.27%) | 86 (14.73%) | 77 (41.40%) | — | — |
| LDL cholesterol (mmol/L) | 4.16 (0.99) | 4.12 (0.97) | 4.15 (1.01) | — | — |
| History of MI (n [%]) | 102 (1.67%) | 9 (1.54%) | 19 (10.22%) | 15 (17.85%) | 3 (10%) |
| History of MI at diagnosis (n [%]) | 79 (25.16) | — | 43 (23.12%) | — | — |
| History of AF at baseline (n [%]) | 58 (0.95%) | 3 (0.51%) | 9 (4.84%) | 33 (39.29%) | 14 (46.67%) |
| History of AF at diagnosis (n [%]) | 145 (44.34%) | — | 87 (46.77%) | — | — |
| History of diabetes (n [%]) | 547 (8.96%) | 53 (9.08%) | 45 (24.32%) | 12 (14.29%) | 3 (10%) |
| Current smoking (n [%]) | 1,620 (27.99%) | 159 (28.24%) | 49 (28.00%) | 7 (8.33%) | 0 |
| Nt-proBNP (pg/mL) | 61.00 (34.00−112.43) | 60.00 (32.14−100.52) | 98.23 (52.14−212.01) | 1036.00 (474.00−2922.00) | 3,997 (2682.75−4929.75) |
| CRP (mg/L) | 1.15 (1.07-1.32) | 1.14 (1.06-1.28) | 1.25 (1.12-1.62) | — | — |
| Creatinine (μmol/L) | 84.76 (16.31) | 84.41 (13.10) | 86.09 (17.19) | 107.14 (34.42) | 117.07 (47.00) |

Characteristics of participants in each of the five study cohorts: (i) the cardiovascular cohort of the population-based Malmö Diet and Cancer Study (MDC-CC), (ii) a random nested subcohort from the MDC-CC for proteomic analysis without prevalent or incident heart failure (HF), (iii) cases with incident heart failure (HF) and plasma available from the MDC-CC, (iv) manifest HF outpatients from the nurse-led HF management program at Skane University Hospital, and (iv) patients with advanced HF who subsequently underwent heart transplantation at Skane University Hospital. Information on a number of parameters was unavailable in the manifest HF cohorts. Continuous variables are presented as mean and standard deviation or median and interquartile range, and categorical variables as count and proportion *BMI* body mass index, *CRP* C-reactive protein, *DBP* diastolic blood pressure, *LDL* low-density lipoprotein cholesterol, *MI* myocardial infarction, *Nt-proBNP* N-terminal pro-B-type natriuretic peptide, *SBP* systolic blood pressure

Supplementary Data 1. Characteristics of the 1305 proteins were examined based on protein annotations in the Human Protein Atlas (HPA, https://www.proteinatlas.org)[13]. In total, 50% of proteins on the assay were annotated as secreted, 30% as intracellular, and the remainder as membrane-bound (Supplementary Fig. 1). The assay is thus enriched for secreted proteins, as only 17% of the 20,981 proteins in the HPA were annotated as secreted and 57% as intracellular. Of all 3617 proteins annotated as secreted in the HPA, 648 (18%) were included on the assay, including the 20 classical plasma proteins that constitute 99% of the total proteome mass because of large molecular mass and low clearance[10]. In terms of tissue-specific expression, 53% showed evidence of tissue enrichment in the HPA whereas 32% were widely expressed.

The representation of established biological pathways by the 1305 proteins was explored based on functional annotations in the biological process component of the Gene Ontology[14] project database (v2.1, accessed on April 12, 2018). Nearly half (5527; 46%) of the total of 11,897 biological processes included at least one of the 1305 proteins on the aptamer-based assay.

**Plasma proteins in HF development**. We first studied the association of plasma proteins with HF development during follow-up in the MDC-CC. To test the association of established biomarkers with incident HF, we fitted Prentice-weighted Cox proportional hazards regression models adjusting for age and sex. The proteins Nt-proBNP, CRP, ST2, troponin T, and TNFα were higher in subjects with incident HF, with similar effects as in previous studies based on conventional immunoassays. Galectin-3 concentration was also nominally higher with similar effect as in previous studies[15–18] (Supplementary Table 3). Of the matrix metalloproteinases (MMPs), only MMP7, MMP12 and none of their tissue inhibitors (TIMPs) were associated with incident HF.

We next tested all proteins on the assay, with the aim to identify novel proteins for HF development. We observed an excess of low p values, with a fivefold higher number of p values below 0.05 than expected under the null hypothesis (354 vs. 65), consistent with several proteins with marked effect (Supplementary Fig. 2a, Supplementary Data 2). A total of 16 proteins passed the Bonferroni-adjusted significance threshold ($p < 3.8 \times 10^{-5}$),

several of which displayed similar effect magnitude as Nt-proBNP (Table 2). To evaluate target binding for the 16 aptamers, we tested correlation with separate immunoassays based on dual antibodies and found significant correlation for all nine proteins with available immunoassays (Spearmans $\rho = 0.37$-$0.77$, $p < 0.05$, Supplementary Fig. 3).

Of the 16 proteins, 12 were secreted and 11 were tissue enriched (Supplementary Table 4). Of individual proteins, the strongest association was observed with increased C5a, a marker of complement system activation. Increased C9, another effector component of the complement system, was also associated with HF. The matricellular protein thrombospondin-2 (TSP2), a regulator of matrix remodeling, has previously been shown to regulate MMP2 [19] and was positively correlated with this protease ($r = 0.17$, $p < 0.001$), but also with MMP7 ($r = 0.18$), MMP12 ($r = 0.24$), MMP13 ($r = 0.13$), MMP17 ($r = 0.18$) and the MMP inhibitor TIMP2 ($r = 0.21$). Conversely, TSP2 was inversely correlated with MMP1 ($r = −0.16$) and TIMP3 ($r = −0.16$). Four of the other proteins were markers of immune system activity (CRP, CXCL13, IL-1RA and IL-18R) and two were involved in plasma coagulation (protein C, tissue plasminogen activator [tPA]). Finally, six of the proteins were intracellular or membrane-bound enzymes: gelsolin, CA13, contactin-1, DUSP3, PRKACA and UNC5H4. The pairwise correlation of the 16 identified proteins is shown in Supplementary Table 5. All identified proteins were only weakly correlated with Nt-proBNP (Pearson's $r \le 0.3$). Most associations were orthogonal to each other, with a few exceptions of biomarker pairs displaying Pearson's $r > 0.5$: C5a was modestly correlated with CRP ($r = 0.58$), whereas CA13, DUSP3 and PRKACA were all strongly correlated ($r > 0.7$). We then tested the independence of identified proteins from established risk factors as described previously[15]. Effect estimates for most proteins remained consistent after adjustment for these factors, but standard errors increased such that only Nt-proBNP, CXCL13 and the complement factors remained significant at the Bonferroni-adjusted threshold (Table 2) although all proteins except IL-1RA and tPA remained nominally significant at $p < 0.05$. Effect estimates declined markedly for CRP, IL-1RA and tPA. Additional adjustment for renal function resulted in only minimal changes in effect estimates (Table 2). In further models of nonischemic HF,

**Table 2 Association of proteins with incident heart failure.**

| | HF, all-cause (age, sex) | HF, all-cause (+risk factors) | HF, all-cause (+renal function) | HF, nonischemic (age, sex) | HF, nonischemic (+risk factors) | HF, nonischemic (+renal function) |
|---|---|---|---|---|---|---|
| **Vascular messengers** | | | | | | |
| Nt-proBNP | $1.84, 4 \times 10^{-7}$ | $1.92, 2 \times 10^{-6}$ | $1.96, 1 \times 10^{-5}$ | $1.62, 2 \times 10^{-4}$ | $1.81, 2 \times 10^{-4}$ | 1.68, 0.002 |
| **Matrix remodeling** | | | | | | |
| TSP2 | $1.55, 8 \times 10^{-8}$ | 1.75, 0.002 | 1.40, 0.01 | $1.52, 9 \times 10^{-7}$ | 1.51, 0.003 | 1.39, 0.02 |
| **Immune system** | | | | | | |
| CXCL13 | $1.45, 6 \times 10^{-8}$ | $1.45, 2 \times 10^{-5}$ | $1.32, 6 \times 10^{-3}$ | $1.44, 3 \times 10^{-7}$ | $1.37, 9 \times 10^{-4}$ | 1.28, 0.03 |
| CRP | $1.86, 3 \times 10^{-7}$ | 1.43, 0.01 | 1.45, 0.01 | $1.67, 7 \times 10^{-5}$ | 1.23, 0.14 | 1.23, 0.02 |
| IL-1R antagonist protein | $1.42, 1 \times 10^{-5}$ | 1.15, 0.26 | 1.16, 0.21 | $1.41, 7 \times 10^{-5}$ | 1.11, 0.46 | 1.16, 0.26 |
| IL-18 receptor 1 | $1.57, 2 \times 10^{-5}$ | 1.45, 0.003 | 1.34, 0.02 | $1.54, 2 \times 10^{-4}$ | 1.37, 0.01 | 1.31, 0.04 |
| **Complement system** | | | | | | |
| C5a | $1.72, 5 \times 10^{-9}$ | $1.67, 3 \times 10^{-5}$ | $1.74, 4 \times 10^{-5}$ | $1.73, 7 \times 10^{-9}$ | $1.56, 9 \times 10^{-5}$ | $1.74, 4 \times 10^{-5}$ |
| C9 | $1.75, 9 \times 10^{-6}$ | $2.08, 2 \times 10^{-6}$ | $1.93, 5 \times 10^{-5}$ | $1.72, 7 \times 10^{-5}$ | $1.91, 2 \times 10^{-5}$ | $2.08, 2 \times 10^{-5}$ |
| **Coagulation system** | | | | | | |
| Protein C | $0.55, 3 \times 10^{-7}$ | $0.64, 4 \times 10^{-4}$ | 0.70, 0.01 | $0.59, 3 \times 10^{-4}$ | 0.67, 0.006 | 0.71, 0.03 |
| tPA | $1.64, 1 \times 10^{-5}$ | 1.21, 0.16 | 1.23, 0.21 | $1.81, 2 \times 10^{-6}$ | 1.49, 0.005 | 1.40, 0.04 |
| **Intracellular or membrane-bound** | | | | | | |
| Gelsolin | $0.55, 6 \times 10^{-8}$ | $0.57, 2 \times 10^{-4}$ | 0.61, 0.001 | $0.53, 4 \times 10^{-7}$ | $0.55, 4 \times 10^{-5}$ | $0.53, 2 \times 10^{-5}$ |
| Carbonic anhydrase 13 | $0.67, 1 \times 10^{-6}$ | 0.77, 0.007 | 0.79, 0.05 | $0.67, 4 \times 10^{-6}$ | 0.78, 0.02 | 0.77, 0.04 |
| Contactin-1 | $0.55, 2 \times 10^{-6}$ | $0.56, 4 \times 10^{-4}$ | 0.63, 0.001 | $0.50, 8 \times 10^{-8}$ | $0.52, 1 \times 10^{-4}$ | $0.54, 2 \times 10^{-5}$ |
| DUSP3 | $0.69, 1 \times 10^{-5}$ | 0.79, 0.02 | 0.82, 0.09 | $0.70, 6 \times 10^{-5}$ | 0.79, 0.01 | 0.80, 0.07 |
| PRKACA | $0.73, 2 \times 10^{-5}$ | 0.80, 0.03 | 0.85, 0.19 | $0.73, 4 \times 10^{-5}$ | 0.82, 0.06 | 0.79, 0.01 |
| Netrin receptor UNC5D | $0.66, 3 \times 10^{-5}$ | 0.78, 0.03 | 0.82, 0.14 | $0.64, 3 \times 10^{-5}$ | 0.79, 0.07 | 0.77, 0.07 |

Proteins associated with heart failure, grouped according to functional annotations and sorted by p value. Results are hazards ratios, scaled to 1 standard deviation for each biomarker, and p values from Cox proportional hazards models with Prentice weighting in patients without heart failure at baseline. The outcome in the first three models is all-cause heart failure and nonischemic heart failure in the final two models. Three models are presented for each outcome, one with adjustment for age and sex, one with additional adjustment for established risk factors for HF, including body mass index, blood pressure (systolic, diastolic, use of antihypertensive therapy), cholesterol concentration (low density and high density), current smoking, history of diabetes and coronary heart disease, and one final model with renal function also as estimated from creatinine

censoring at incident myocardial infarction when preceding HF diagnosis, effect estimates remained consistent except for CRP which further declined.

We next compared concentrations of the 16 plasma proteins in the cohort of outpatients with manifest HF with the general population cohort from the MDCS. All proteins, except the Netrin receptor UNC5D, were markedly different in HF patients from the general population after adjustment for age, sex and BMI in logistic regression models in a direction consistent with that for incident HF cases but more pronounced (Fig. 2, Supplementary Table 4 and Supplementary Fig. 4).

**Complement and coagulation systems in HF development.** The broad coverage offered by the aptamer array allowed detailed interrogation of many components of the well-defined complement and coagulation systems. The strong association of C5a and C9 with HF and enrichment of complement activation pathways suggests activation of the complement cascade in subjects at risk of HF. We observed consistent directionality and low p values for all components of the effector and amplification stages of the complement system (Fig. 3), with increased C3a, C5a, C5b-C6 and other components of the terminal complement complex (C7, C8, C9). No association was observed with components in the classical and lectin-mediated activation pathways. Higher levels of mediators in the alternative pathway were observed (C3, B), of C3b breakdown products (iC3b, C3d) and factors mediating C3b breakdown (I, H) whereas DAF, a negative regulator of C3bBb generation, was lower, collectively consistent with increased activity in the alternative pathway.

For the coagulation system, we first explored the influence of warfarin, the only oral anticoagulant available at the time of the MDCS baseline exam, which acts to reduce concentrations of coagulation factors II, VII, IX, and X, as well as proteins C and S. Warfarin may confound the association of coagulation markers with incident HF, as it is commonly used in subjects with atrial

fibrillation (AF), and AF is often the first manifestation of HF[20]. Only seven subjects reported use of warfarin at baseline, all of whom were incident HF cases. We confirmed markedly lower levels of all six proteins in warfarin users at baseline (all $p < 0.05$). Exclusion of warfarin users greatly attenuated the association with protein C (HR 0.66, $p = 0.002$), but not with tPA (HR 1.56, $p = 1 \times 10^{-4}$). In detailed interrogation of the coagulation system, associations were also observed with lower antithrombin III, higher factor IX, TFPI (Fig. 4), von Willebrand factor (central for platelet activation, not shown in figure), and nominally with higher tissue factor and lower factor VII. Thus, although the assay was unable to distinguish the proenzyme (zymogen) forms of most coagulation proteins from the activated enzymes, results indicated perturbation of all the key coagulation regulatory proteins (tPA, TFPI, antithrombin III, protein C) and suggested increased activity through tissue factor, the principal activator of coagulation under physiological conditions.

**Plasma proteins in manifest HF.** We next explored the association of biomarkers with manifest HF, as compared to the general population in the MDC-CC subset using logistic regression models adjusted for age and sex. Consistent with previous studies, HF patients had higher circulating levels of Nt-proBNP, CRP, ST2, galectin-3, troponin T, tumor necrosis factor (TNF) α, and renin[3]. Several MMPs and TIMPs were also markedly increased in HF (Supplementary Table 3). The small peptide hormone angiotensin II, steroid hormone aldosterone and the cathecholamines were not measurable by this assay.

We next tested all 1305 aptamer-targeted proteins for association with HF. We observed a marked excess of low p values, with a 13-fold higher number of p values below 0.05 than expected under the null hypothesis (842 vs. 65), consistent with many proteins of marked difference in concentration (Supplementary Fig. 2b, Supplementary Data 3). A total of 421 proteins (32%) passed the Bonferroni-adjusted significance

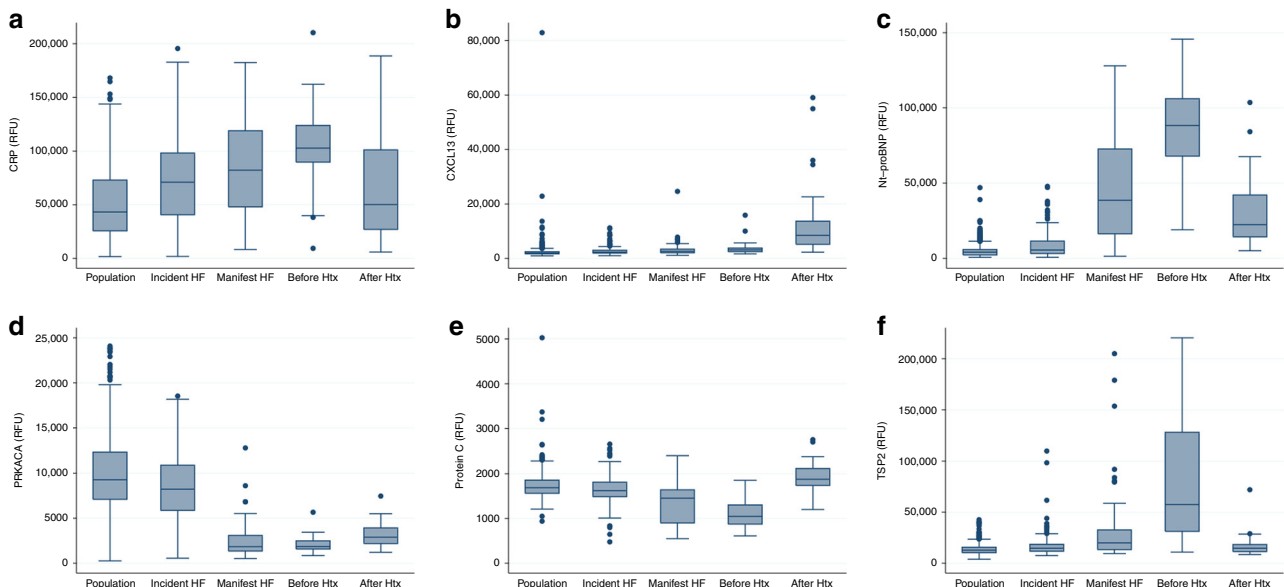

**Fig. 2 Plasma protein abundance across heart failure stages.** Box plots depicting distribution of (**a**) CRP, (**b**) CXCL13, (**c**) Nt-proBNP, (**d**) PRKACA, (**e**) Protein C, and (**f**) TSP2 across cohorts representing different stages of heart failure. In addition to association with incident heart failure, these six proteins changed markedly (>50%) after heart transplantation (HTx). Cohorts include individual human subjects from a population-based cohort who did not develop heart failure during follow-up ("Population", $n = 583$), from the population-based cohort who did develop heart failure ("Incident HF", $n = 185$), subjects with manifest heart failure ("Manifest HF", $n = 84$), and serial measures from subjects with advanced heart failure before and after heart transplantation ("Before Htx" and "After Htx", $n = 30$ for both). All except CXCL13 changed in directional consistency with the risk estimate for incident HF towards the general population. The box center line indicates the median, box bounds represent 25th and 75th percentiles, and the whiskers are the most extreme values within 1.5× the interquartile range from the nearer quartile. RFU relative fluorescence units.

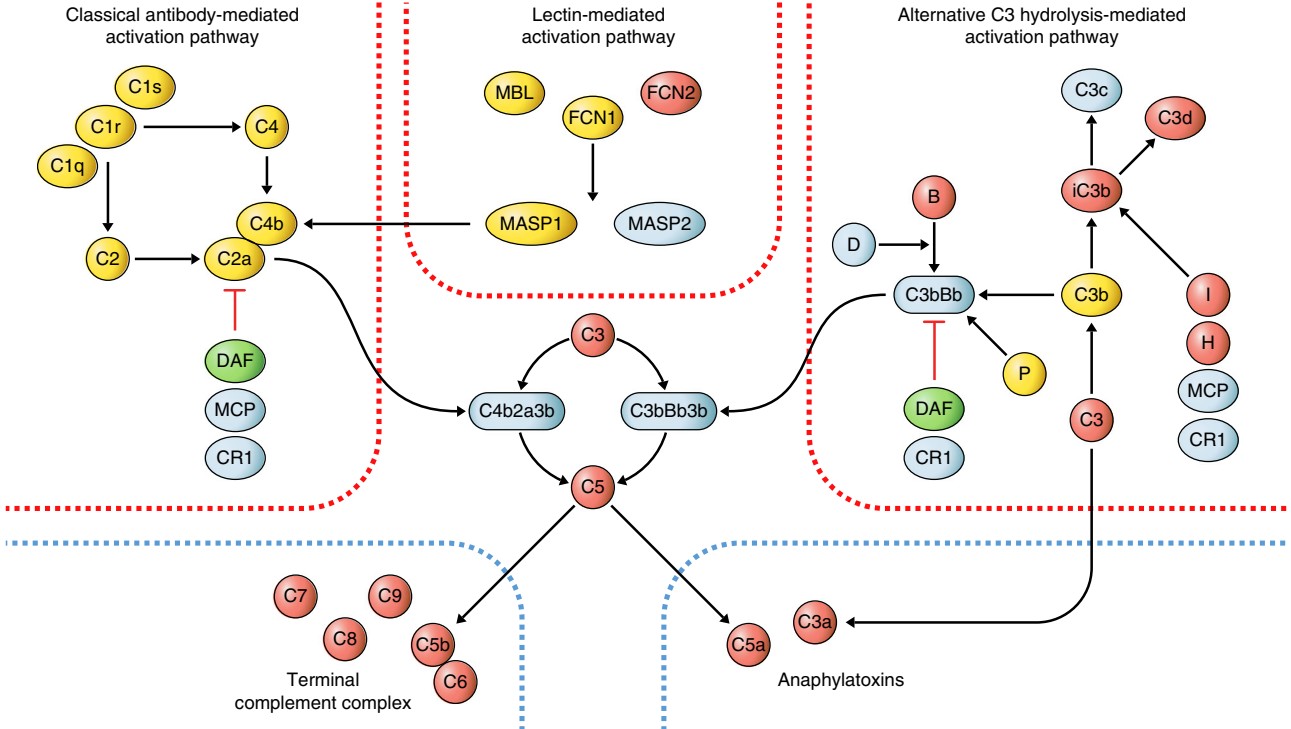

**Fig. 3 Association of complement system components with development of heart failure.** Diagram of protein interactions in the complement system based on a published review[59]. The three activation pathways (red borders) lead to activation of complement factor 3 which amplifies the signal via feedback loops, and further activates the two common effector pathways: the anaphylatoxins and the terminal complement complex (blue dashed borders). Directionality of significant associations ($p < 0.05$) with incident heart failure (HF) from Cox proportional hazards regression models adjusting for age and sex ($n = 583$ population-representative controls, 185 cases) is indicated for each component with color codes, where red indicates higher concentration with incident HF, green lower concentrations, and yellow no significant association. White color indicates that the component was not present on the aptamer-based proteomics platform.

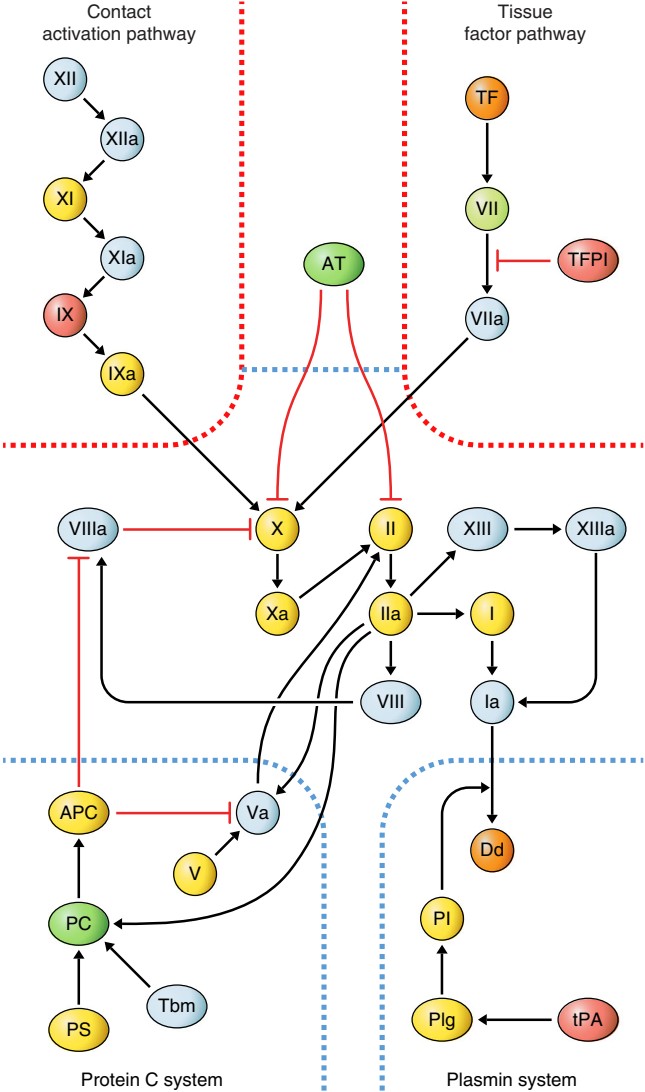

**Fig. 4 Association of coagulation system components with development of heart failure.** Diagram of protein interactions in the coagulation system based on a published review[60]. The two activation pathways (red borders) lead to activation of factor X which further activates II (thrombin) and subsequently fibrin and XIII, which form the bloodclot. The bloodclot is digested into d-dimers by the plasmin system, and three additional systems (blue dashed borders) negatively regulate clotting (protein C, AT, TFPI). Directionality of nominally significant associations ($p < 0.05$) with incident heart failure (HF) from Cox proportional hazards regression models adjusting for age and sex ($n = 583$ population-representative controls, 185 cases) is indicated for each component with color codes, where red indicates higher concentration with incident HF, green lower concentrations, and yellow no significant association. Nominally significant associations ($p < 0.10$) are indicated in orange for higher (two components) and mint green for lower (one component) concentration. All associations were examined only after exclusion of subjects on oral anticoagulation therapy (warfarin). White color indicates that the component was not present on the aptamer-based proteomics platform. APC activated protein C, AT antithrombin III, Dd d-dimer, PC protein C, Pl plasmin, Plg plasminogen, PS protein S, Tbm thrombomodulin, TFPI tissue factor pathway inhibitor, TF tissue factor.

threshold ($p < 3.8 \times 10^{-5}$, Supplementary Data 4). Of note, half of these proteins (56%) were lower in HF cases, of which most (74%) were annotated as intracellular or membrane-bound proteins and were widely expressed (61% expressed in all HPA

tissues) (Supplementary Fig. 1) although several were of likely vascular (P-selectin, tropomyosin-4) or skeletal muscle (fibroblast growth factor 6) origin. In contrast, 61% of the proteins that were higher in HF cases showed evidence of tissue enrichment and 68% were secreted proteins, including several well-known proteins with a highly plausible role in HF pathophysiology such as vascular endothelial growth factor A, angiogenin, endostatin, glucagon, erythropoietin, periostin. Additional adjustments did not markedly change the results (Supplementary Data 3). To study the external validity of this observation, we compared our results with an independent set of HF cases and controls (BioVU study) and observed significant correlation of odds ratios for the 352 of the significant proteins that were available in both cohorts (Pearson's $r = 0.54$) and consistent effect directionality in 75%.

To test for enrichment of established biological pathways among plasma proteins associated with manifest HF, gene-set enrichment analysis was applied to the 421 proteins associated with HF. A total of 43 of the 50 hallmark gene sets studied were enriched among the proteins (FDR < 0.05%) as shown in Supplementary Table 6. The most strongly enriched gene sets were the PI3K/AKT/mTOR pathway, the complement system, and the epithelial−mesenchymal transition pathway active in wound healing and fibrosis, but also included gene sets representing inflammation, apoptosis, fibrosis, and metabolism.

**Plasma proteins and reversal of HF state**. To further test the association of proteins with HF as opposed to comorbidities, we compared serial measurements from 30 advanced HF patients before and 6 months after heart transplantation. Distribution of the 16 proteins associated with HF development before transplantation were comparable to the HF outpatient cohort, although many proteins displayed marked further deviation from the general population (Fig. 2 and Table 3). A total of 12 of the 16 proteins (75%) changed in directional consistency with the risk estimate for incident HF towards the general population, supporting origin in the heart or in other tissues in response to HF, rather than comorbid conditions. Of the other four proteins, CXCL13 increased markedly, contactin-1 decreased, tPA increased and DUS3 did not change. Of the 421 proteins associated with manifest HF, 158 (38%) changed significantly after heart transplantation ($p < 0.05$) towards levels observed in the general population (Supplementary Data 4). All 138 proteins that changed after transplantation ($p < 3.8 \times 10^{-5}$) are presented in Supplementary Data 5.

**Tissue and cellular origin of plasma proteins**. We next explored the likelihood of cardiac origin, as opposed to origin from other tissues as a consequence of HF. First, we compared protein concentrations in coronary sinus blood to peripheral venous blood in six patients with hypertrophic obstructive cardiomyopathy obtained before undergoing alcohol septal ablation. Numerically higher coronary sinus level was observed in 6 of 16 proteins (C5a, Gelsolin, TSP2, tPA, IL-18R, Netrin receptor UNC5D) (Supplementary Table 7). Second, we examined cardiac expression in publically available datasets: gene expression in the Genotype-Tissue Expression (GTEx), HPA, and Tabula Muris projects[13,21,22], and protein expression from immunohistochemistry patterns in the HPA and a recent cardiac mass spectrometry dataset[23]. Results from the publically available datasets were largely concordant with the coronary sinus enrichment data. Consistent evidence for cardiac expression and enrichment in cardiac blood was observed for Nt-proBNP, gelsolin, TSP2, and IL-18R. Three of the other proteins were only expressed in liver (CRP, C9, protein C) as was the C5 protein, while other proteins were more widely expressed.

**Table 3 Dynamics of proteins for heart failure development before and 6 months after heart transplantation.**

| Protein | Before | After | Change | *p* value |
|---|---|---|---|---|
| Vascular messengers | | | | |
| Nt-proBNP | 88,415 (69,500−105,996) | 22,494 (14,353−41,738) | −75% | <0.001 |
| Matrix remodeling | | | | |
| TSP2 | 57,588 (31,334−127,082) | 14,626 (11,767−18,097) | −75% | <0.001 |
| Immune system | | | | |
| CXCL13 | 3194 (2477−3866) | 8482 (5180−13,676) | +166% | <0.001 |
| CRP | 102,824 (89,798−124,008) | 50,247 (27,082−101,158) | −51% | 0.01 |
| IL-1R antagonist protein | 5073 (3997−6947) | 4858 (3911−5456) | −4% | 0.13 |
| IL-18 receptor 1 | 12,489 (9524−16,322) | 12,032 (9441−15,388) | −4% | 0.47 |
| Complement system | | | | |
| C5a | 14,538 (11,751−17,956) | 13,636 (10,065−19,109) | −6% | 0.99 |
| C9 | 86,201 (79,232−95,702) | 80,443 (70,331−95,042) | −7% | 0.28 |
| Coagulation system | | | | |
| Protein C | 1048 (874−1304) | 1874 (1737−2114) | +79% | <0.001 |
| tPA | 416 (363−584) | 475 (325−644) | +14% | 0.38 |
| Intracellular or membrane-bound | | | | |
| Gelsolin | 700 (654−800) | 868 (800−950) | +24% | <0.001 |
| Carbonic anhydrase 13 | 1156 (543−1500) | 1485 (877−2219) | +28% | 0.06 |
| Contactin-1 | 262 (224−294) | 198 (180−235) | −24% | <0.001 |
| DUSP3 | 1215 (912−1926) | 1213 (875−2115) | 0% | 0.44 |
| PRKACA | 1874 (1589−2486) | 2888 (2189−3923) | +54% | 0.009 |
| Netrin receptor UNC5D | 4849 (3655−5617) | 5875 (4853−7490) | +21% | <0.001 |

Protein levels before and after transplant values presented as median (25th−75th percentile), based on serial measures from 30 heart transplant recipients. Abundance estimates were compared with the Wilcoxon signed-rank test. Abundance expressed as relative fluorescence units (RFU). Proteins grouped according to functional annotations

To explore the cellular origin of the four proteins with evidence of cardiac origin, we performed spatial transcriptomic analysis of heart samples from two subjects with heart failure[24]. We observed patchy expression of Nt-proBNP, TSP2 and gelsolin (Supplementary Fig. 5a). The expression of IL18-R was below the level of detection and was not included, as was the presence of leukocytes as determined by *PTPRC* (CD45) expression. The heart mainly consists of three cell types: cardiomyocytes, endothelial cells and fibroblasts, as described previously[25]. Spots on the tissue slide with detectable Nt-proBNP displayed coexpression with *TNNT2* and *TNNI3*, indicating cardiomyocyte origin, whereas gelsolin displayed coexpression primarily with *PECAM1* and *MYH11*, indicating vascular origin (Supplementary Fig. 5b). Myocardial TSP2 expression was lower, but tended to colocalize with *VIM* and *POSTN*, suggestive of a primary fibroblast origin. Generalizability of these findings should be cautious, as based on only two samples, but myocardial expression of NT-proBNP is well established and corroborating fibroblast expression for TSP2 was observed in cell lines from the HPA and mouse cardiac single-cell transcriptomes from the Tabula Muris project (Supplementary Fig. 6).

**Genome-wide association studies of plasma proteins.** Finally, to further explore the specificity of aptamer reagents and implicated pathways for the 16 proteins associated with incident HF, we conducted genome-wide association studies (GWAS) of these proteins in 1421 subjects from the MDCS representing the case-cohort subset described above and additional case subsets with cardiometabolic disease not contributing to the current study. Quantile−quantile plots for the 16 GWASs did not indicate genomic inflation (Supplementary Fig. 7). Genome-wide significant associations (protein quantitative trait loci [pQTLs], $3 \times 10^{-9}$) at 11 loci were observed for nine proteins (Table 4, Supplementary Table 8 and Supplementary Fig. 8). In a second stage, we expanded the analysis of *cis*-regulatory variants by meta-analysis with 759 subjects from a previously published dataset[26], as variants in *cis* provide important information that support target binding for affinity reagents[10]. This analysis resulted in the discovery of loci in *cis* for five proteins ($p < 2 \times 10^{-5}$). The proportion of protein variability explained was modest for most SNPs (1−6%, Table 4), but higher for IL-18R1 (37%), TSP2 (13%), and protein C (15%). pQTLs were identified for all of the ten secreted proteins except CXCL13 and for four of the six intracellular proteins. pQTLs in *cis* (located within 1 Mb of the transcription start site for the gene encoding the protein under study; Fig. 5) were observed for 13 of the proteins and in *trans* for 3 (C5a, contactin-1, protein C; Fig. 5). For C5a, one locus in *cis* and one in *trans* were identified, of which the latter encodes complement factor H, underscoring the key role of factor H for inactivation of C3 in the alternative pathway, resulting in lower C5a generation. Other pQTLs in *trans* included for protein C the locus with the endothelial protein C receptor which is central for protein C activation, and one locus for contactin-1. Five of the pQTL loci in *cis* (Ile44Thr in *IL1F10* for IL1RN, Pro167Ser in *C9* for C9, Arg164Trp in *PLAT* for tPA, Ala78Thr in *GSN* for gelsolin, Gly18Ala in *E2F5* for Carbonic anhydrase 13) contained known missense variants at $r^2 > 0.8$ in HapMap III CEU, as did two in *trans* (Ser219Gly in *PROCR* and Ala25Thr in *MYH7B* for protein C, Ile310Val in *TMEM8A* for contactin-1). Six groups have recently reported GWAS of different versions of the aptamer-based assay in different cohorts[11,26–30], with our cohort representing the third largest to date and with the third largest number of aptamers. Of the identified loci, the C5a locus in *trans* and the loci for contactin-1 and gelsolin were novel, whereas 13 had been described previously[11,26,29,30]. A previous study also identified a pQTL in *trans* for PRKACA[11] which however did not pass the significance threshold in our study.

We explored association with gene expression for index SNPs at each locus in publically available eQTL datasets using PhenoScanner[31]. Of the 13 pQTLs in *cis*, 10 were also eQTLs (Table 4), including three in heart tissue (Nt-proBNP, TSP2, IL-18R). All pQTLs in *trans* were also eQTLs: one C5a locus (rs485632) was associated with complement factor H expression, the protein C locus with the endothelial protein C receptor, and the contactin-1 locus with *TMEM8A*.

**Table 4 Genetic associations with plasma proteins.**

| Protein | Locus (p value) | Location | Effect | Gene expression |
|---|---|---|---|---|
| **Vascular messengers** | | | | |
| Nt-proBNP | 1p36 ($5 \times 10^{-9}$)* | cis | 2% | NPPB (heart) |
| **Matrix remodeling** | | | | |
| TSP2 | 6q27 ($2 \times 10^{-47}$) | cis | 13% | THBS2 (heart) |
| **Immune system** | | | | |
| CXCL13 | — | — | — | — |
| CRP | 1q23 ($6 \times 10^{-10}$) | cis | 2% | — |
| IL-1R antagonist protein | 2q13 ($9 \times 10^{-17}$) | cis | 5% | IL1RN (multiple) |
| IL-18 receptor 1 | 2q12 ($9 \times 10^{-135}$) | cis | 37% | IL18R1 (heart) |
| **Complement system** | | | | |
| C5a | 9q33 ($3 \times 10^{-10}$) * | cis | 1% | C5 (multiple) |
| | 1q31 ($2 \times 10^{-10}$) | trans | 3% | CFH (multiple) |
| C9 | 5p13 ($4 \times 10^{-10}$) | cis | 5% | — |
| **Coagulation system** | | | | |
| Protein C | 2q14 ($2 \times 10^{-19}$) | cis | 6% | PROC (liver, atria) |
| | 20q11 ($2 \times 10^{-57}$) | trans | 15% | PROCR (multiple) |
| tPA | 8p11 ($1 \times 10^{-7}$) * | cis | 1% | PLAT (multiple) |
| **Intracellular or membrane-bound** | | | | |
| Gelsolin | 9q33 ($2 \times 10^{-9}$) * | cis | 1% | GSN (blood) |
| Carbonic anhydrase 13 | 8q21 ($2 \times 10^{-19}$) | cis | 6% | CA13 (multiple) |
| Contactin-1 | 12q12 ($2 \times 10^{-11}$) | cis | 3% | CNTN1 (multiple) |
| | 16p13 ($1 \times 10^{-17}$) | trans | 5% | TMEM8A (multiple) |
| DUSP3 | — | — | — | — |
| PRKACA | — | — | — | — |
| Netrin receptor UNC5D | 8p12 ($1 \times 10^{-6}$) * | cis | 1% | — |

Index SNPs at loci associated with the 16 heart failure proteins at genome-wide significance ($p < 5 \times 10^{-8}$). p values from linear regression models are presented for associations in cis (within 500 kb) and trans. Effects of each SNP are expressed as proportion of variability explained. Association of index SNPs at each locus with gene expression was explored in publically available eQTL data using PhenoScanner[31], with results presented for transcripts below a Bonferroni-corrected threshold for the total nr of SNPs ($p < 0.05/15$). Proteins associated with heart failure, grouped according to functional annotations. *Asterisk indicates that the locus was discovered in the second stage of discovery, in a meta-analysis of two cohorts focused on cis-windows (±500 kb from the transcription start site), with a lower number of SNPs tested in each window, resulting in a more permissive significance threshold ($2 \times 10^{-5}$)

## Discussion

We have characterized the plasma proteome of human subjects with HF in different stages. Our findings are consistent with activation of a number of pathways in early stages of HF, including immunological, complement, coagulation, natriuretic and extracellular matrix remodeling pathways, whereas pervasive perturbation of the plasma proteome was observed in manifest HF. Activity in several of these pathways declined after heart transplantation, indicating that protein perturbations were induced by the failing heart rather than comorbid conditions. Consistent evidence for a direct cardiac origin was only observed for a subset of proteins, including Nt-proBNP, interleukin-18 receptor, thrombospondin-2, and gelsolin, while the activated form (C5a) of the liver-specific protein C5 was also enriched in coronary blood. Cardiac spatial transcriptomics indicated a cardiomyocyte origin for Nt-proBNP, fibroblast origin for thrombospondin-2 and vascular origin for gelsolin.

The interpretation of any biomarker association with HF is complicated by the heterogeneity of this condition, resulting from a wide range of processes negatively influencing the heart and in itself influencing multiple organ systems. This study was therefore designed to study molecular patterns of HF both in early and advanced stages of HF, as well as after reversal of HF with heart transplantation. Molecular patterns consistently associated with HF in both early and advanced stages are more likely to represent mediators of HF rather than consequences. Patterns that can be reversed with transplantation are unlikely to represent comorbid conditions. Moreover, we performed extensive adjustments for comorbid conditions to assess generalizability and stratified HF according to history of myocardial infarction in sensitivity analyses, on the basis of myocardial infarction being a clearly distinguishable etiology with a primarily vascular pathophysiology—a strategy that we and others recently validated to improve yield

of biological insights[32]. The large number of proteins associated with manifest HF are of great interest, especially the secreted proteins that were higher in HF subjects, but will need to be replicated in independent studies and evaluated for relation to outcomes in larger studies. Here, we focused on the pathways associated with early HF stages, all of which were also further pronounced in subjects with manifest and advanced HF. Although the consistent association across multiple stages of HF provide for some reassurance of the validity of these findings, external replication of these findings is also warranted. We note that most of these proteins were consistently associated with nonischemic HF, although significantly reduced associations were observed for CRP and IL-1RA, both previously associated with coronary disease[33,34], suggestive that these associations may be mediated by coronary disease or its risk factors. It is also important to note that this study was designed to explore common molecular patterns of HF and was underpowered to study etiological or morphological subgroups of HF. In practice however, specific etiologies are often difficult to determine, and morphological characteristics are subject to temporal variability[35].

The broad pathways identified for HF development have been implicated previously[3], and we confirmed association with most established markers. However, the markers identified here may better capture the information content of the implicated pathways and could constitute therapeutic targets. First, we identified immunological mediators, where CRP and TNFα inhibitors have been consistently associated with HF, including in the current study, but a causal role of CRP has been disproven in genetic studies[34] and therapeutic TNFα inhibition failed in clinical trials[36]. Promising data have however emerged from systematic genome-wide screens for other immunological mediators, such as cardiac alarmins IL-33[37] and TSLP[38], and animal studies have

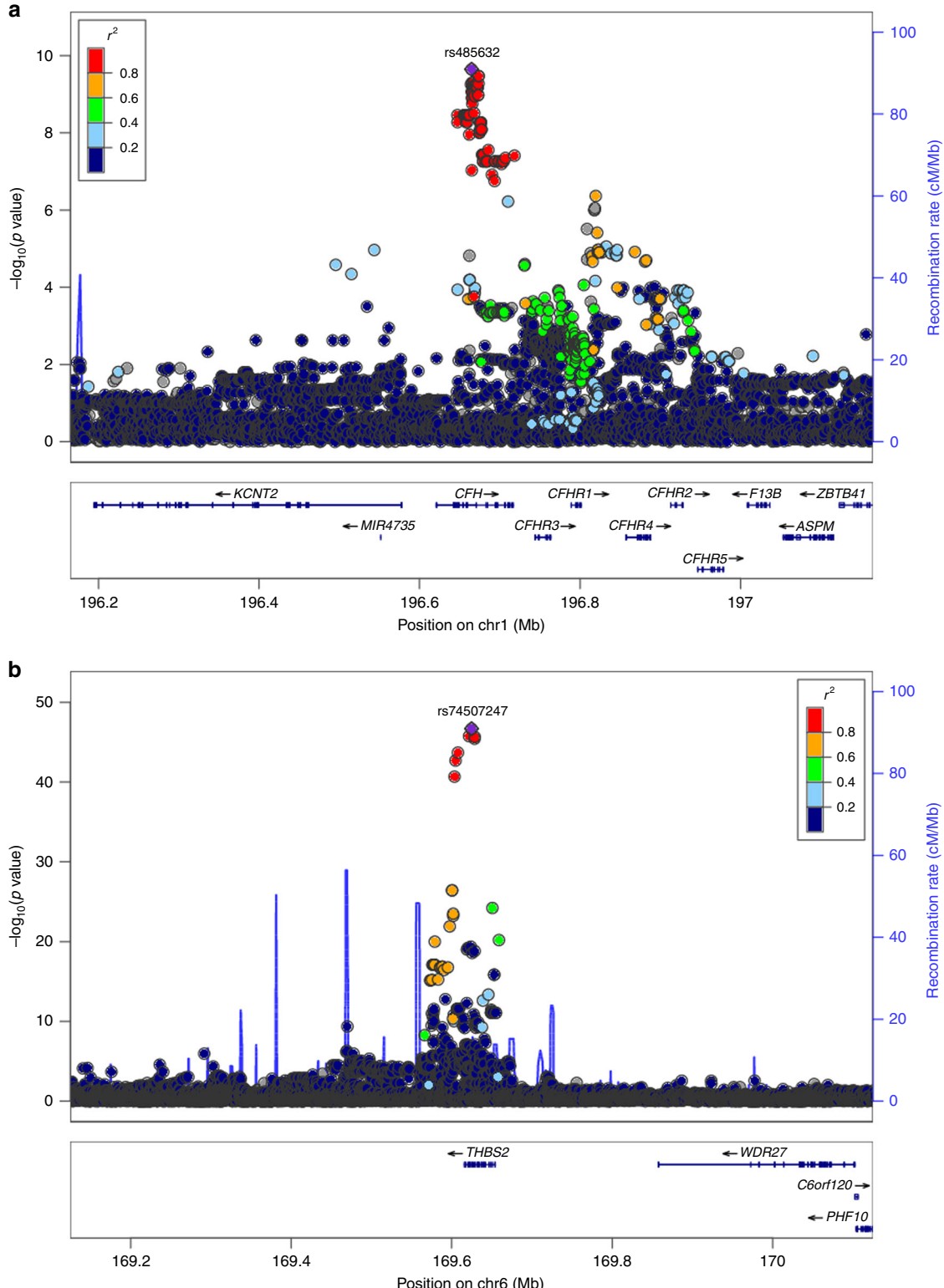

**Fig. 5 Association of regulatory genetic variants in *cis* and *trans* with plasma proteins.** Regional association plots of genetic loci in **a** *cis*-associated with thrombospondin-2, and **b** in *trans*-associated with C5a at the *CFH* locus. Plots depict all single nucleotide polymorphisms (SNPs) within ±500 kb of the lead SNP. Circles indicate SNPs with genomic position on the *x* axis (human genome build 19), *p* value on the left-hand *y* axis and recombination rate on the right-hand *y* axis. Colors indicate pairwise linkage disequilibrium of each SNP with the lead SNP at each locus.

implicated regulators of both T-cell[39] and B-cell activity[40]. Our results further nominate the cytokines IL-18 and CXCL13 which activate and recruit T and B cells, respectively. A form of the receptor for IL-18 circulating at higher concentrations showed consistent evidence for cardiac origin, and is thought to act as a decoy for IL-18 to limit systemic leakage as also seen for IL-33 with the ST2 receptor. Genetic variants associated with IL-18R in *cis* were also associated with cardiac expression, providing further

support for cardiac origin. For CXCL13, our results were consistent with extracardiac release, and rather than reduction after transplantation we observed a striking increase. Current immunosuppression in heart transplantation targets T lymphocytes, but there is much evidence indicating that B-cell activation results in an antibody-mediated form of rejection and contributes to development of transplant vasculopathy, the leading cause of graft failure after transplantation[41]. Further study of CXCL13 is thus warranted also in the transplantation context.

Second, we identified a marker of extracellular matrix remodeling, a system which has been studied previously in HF, particularly the MMPs and TIMPs. We identified thrombospondin-2, a matricellular protein that mediates cell−matrix interactions, to be robustly associated with both HF development and manifest HF, decline markedly after transplantation, and with consistent evidence of cardiac origin. Genetic variants in *cis* associated with thrombospondin-2 were also associated with cardiac gene expression, and a distinct correlation pattern was observed with MMPs and TIMPs. Thrombospondin-2 appears to be a good therapeutic candidate for prevention of adverse remodeling, which has previously been implicated through gene expression microarrays of murine HF models as a regulator of myocardial fibrosis, repair, and MMPs[19,42–44].

Third, we identified the well-defined and protein-based complement and coagulation systems which, in contrast to the complex immune and matrix remodeling systems, have received less attention in the context of HF. The broad coverage of the aptamer-based platform allowed detailed investigation of the protein components of these pathways. In the complement system, sequential activation of complement proteins results in signal amplification, such that a small increase in early factors can result in a substantial increase in the signaling effectors, C3a, C3b, C5a and C5b, which recruit inflammatory cells, and the terminal complement complex (TCC), C5b-C9, which mediates lysis of bacteria. We observed a marked increase of all effector proteins in subjects with incident HF, which along with higher levels in coronary sinus blood than peripheral blood and a slight decline after transplantation, indicates cardiac complement activation in HF development. We also observed a more modest increase of proteins specifically in the alternative activation pathway in patients with incident HF. These observations corroborate a few previous reports that have implicated activation of the alternative pathway in manifest HF[45], and may be particularly important as recent work has pointed to a central role of C5a in myocardial repair and regeneration[46,47].

For the coagulation system, we observed increased activity of the regulatory systems including tPA, protein C, TFPI and antithrombin, and some evidence for tissue factor pathway activation, the principal activator of coagulation. Detailed characterization of the prognostic effects of these proteins in HF may be of clinical relevance to identify subgroups who could benefit from anticoagulation, as risk of thromboembolism is increased in HF but the overall risks of anticoagulation outweigh the benefits[48].

Finally, we also identified six intracellular or membrane-bound proteins. All displayed inverse associations with HF development were markedly lower in manifest HF, and most increased after transplantation. Circulating concentrations were lower in coronary sinus than in peripheral blood, indicating a predominant non-cardiac origin for all except gelsolin where spatial transcriptomics indicated a vascular origin. The three of the proteins that are cytosolic were strongly correlated (carbonic anhydrase 13, DUSP3 and PRKACA) and were all widely expressed. Interestingly, in manifest HF patients lower concentrations were also observed for a large number of predicted intracellular proteins with widespread expression, which improved after transplantation. The mechanism

underlying these observations remains unclear, but could reflect hypoperfusion in HF of tissues with active exosomal signaling or persistent protein leakage rather than downregulated potentially protective pathways.

A potential limitation of the current study is that aptamers, like all affinity reagents, are difficult to validate with regard to binding specificity. However, extensive assessment of cross-reactivity with homologous proteins has been conducted previously[11], with limited off-target reactivity observed for most proteins and no cross-reactivity observed for any of the 16 proteins associated with HF development. We here describe correlation with dual-binder immunoassays for all nine of the 16 proteins that were available on both platforms. We also applied a genetic approach to study target binding, confirming identification of genetic variants in *cis*, i.e. near the gene encoding the protein of interest, for 13 of the 16 proteins associated with HF development. Six previous studies have also confirmed associations in *cis* for 11 of the proteins[11,26–30]. Another potential limitation is the impact of preanalytical factors. Of preanalytical factors, sampling conditions, sample tubes and biobanking conditions were standardized in each of the cohorts. However, for the comparison of MDCS with manifest HF cases, two preanalytical factors could potentially contribute: the lack of standardized fasting conditions in the HF cohort, and the longer storage times for the MDCS. Although limited impact of fasting has been described with the Somascan assay used here, effects of long-term storage remain unclear. However, the largely consistent effect directionality and effect correlation with the independent BioVU cohort argue against a major impact of preanalytical factors on the results. Blood samples were obtained from peripheral veins in the population-based and manifest HF settings, but from a central venous catheter in the heart transplant cohort. It is well established that proteases and other clearance mechanisms may rapidly degrade certain proteins with mainly local effects, which could result in substantial abundance differences across vascular territories. Although many proteins were further perturbed with directional consistency in the heart transplant candidates as compared to the manifest HF cohort, we recognize that the impact of differences between vascular beds for the proteins studied here remains unclear. We also recognize that the patients from whom coronary sinus samples were drawn all represented a more distinct pathophysiological entity, hypertrophic obstructive cardiomyopathy, that may not in all aspects generalize to the heart failure in general. Finally, we also recognize that some extent of unmeasured confounding that we were unable to account for cannot be ruled out, as in all observational studies.

In summary, this study demonstrates the feasibility of systematic plasma biomarker discovery for HF and provides a comprehensive view of the plasma proteome across populations at risk of HF, with manifest HF, and heart transplant recipients. Our findings demonstrate broad perturbations in manifest HF and implicate several proteins associated with HF development, including natriuretic peptides, immunological mediators, matrix turnover mediators, the alternative complement pathway, the coagulation cascade, and several intracellular proteins, highlighting the contributions of multiple organ systems to the complex HF condition. Future mechanistic studies will be needed to determine any therapeutic potential of these proteins and pathways, and large clinical cohorts will be needed to evaluate the ability of these markers to distinguish disease subsets with variable outcome towards precision medicine for HF.

## Methods

**Study samples.** Samples from three cohorts underwent aptamer-based proteomic analysis as an initial stage of the European Research Council-funded research program "Integrative omics of heart failure to inform discovery of novel drug

targets and clinical biomarkers" (inHForm)[5]: (1) the population-based MDC-CC cohort, (2) a hospital-based cohort with manifest HF, (3) serial samples from a cohort of patients undergoing heart transplantation. In addition, we sought validation of associations with manifest HF in the bioVU cohort, and evidence of cardiac origin for proteins in dual samples from coronary sinus and femoral vein of patients undergoing alcohol septal ablation for hypertrophic obstructive cardiomyopathy. Study in all cohorts complied with all relevant ethical regulations, was approved by the local ethics committees (at Lund University, Massachusetts General Hospital, or Vanderbilt University Medical Center), and all participants provided informed consent.

The Malmö Diet and Cancer Study is a prospective cohort study, which includes 30,447 men (born between 1923 and 1945) and women (born between 1923 and 1950) from the city of Malmö in southern Sweden. Participants underwent baseline examinations between 1991 and 1996, including anthropometric measures, blood pressure measurement using a mercury-column sphygmomanometer after 10 min of rest in the supine position, and filled out a questionnaire[49]. From this cohort 6103 individuals with a baseline examination between 1991 and 1994 were randomly selected to participate in a study of cardiovascular risk factors, the MDCS Cardiovascular Cohort (MDC-CC), of whom 5543 underwent blood sampling from the median cubital vein in ethylenediaminetetraacetic acid (EDTA)-coated tubes under standardized fasting conditions[15]. Samples were immediately centrifuged and plasma extracted, aliquoted and stored at −80 °C. Data on current smoking, diabetes mellitus and use of antihypertensive and antidiabetic medications was ascertained from a questionnaire. Diabetes mellitus was defined as fasting blood glucose >6.0 mmol/L, self-reported physician diagnosis or use of antidiabetic medications. For the current study, a random subset was selected from the MDC-CC for proteomic profiling ($n = 583$). The baseline distribution of clinical characteristics in the MDC-CC random subcohort was similar to the full MDC-CC, as shown in Table 1. Subjects with incident HF during follow-up ($n = 185$) were identified from nation-wide registers with high case validity as described previously[15].

For manifest heart failure, peripheral venous blood samples were collected in EDTA-coated tubes through median cubital vein venipuncture from 84 patients attending an outpatient-based, nurse-led heart failure management program in 2013−2014, as part of an ongoing biobanking project at Skåne University Hospital (Lund Cardiopulmonary Register)[35]. Fasting conditions were not standardized. Samples were immediately centrifuged at 2000 × g for 10 min and plasma extracted, aliquoted and stored at −80 °C until the time of biochemical analysis for the current study without previous thawing.

For heart transplant recipients, venous plasma samples were drawn serially from central venous catheters for 30 patients with advanced heart failure undergoing right heart catheterization between 2012 and 2016 as part of the routine assessment of eligibility for heart transplantation at Skåne University Hospital in Lund, Sweden. Sample treatment and biobanking was identical to the procedure for the advanced HF patients, and there was no patient overlap between the cohorts. Only subjects living in the secondary care catchment area of Skåne University Hospital and thus scheduled for follow-up at this institution were considered for study inclusion. Follow-up samples were obtained from the same patients at the time of the 6-month follow-up right heart catheterization with endomyocardial biopsy after heart transplantation. Baseline samples were collected at a median of 99 days before transplantation (range 4−983) and follow-up samples at a median of 167 days after heart transplantation (range 149−219). No patient had evidence of graft rejection by three endomyocardial biopsies obtained simultaneously.

For validation of associations with manifest HF, residual blood samples from routine clinical care that would otherwise be discarded were collected between September 2014 and September 2015 within BioVU, the Vanderbilt University Medical Center biorepository. Samples were obtained from 48 HF cases and 47 controls without HF or cardiovascular disease. The samples were robotically processed for plasma extraction and stored at −80 °C. The Vanderbilt Institutional Review Board approved this study and all subjects consented to participate in BioVU.

For evidence of cardiac origin, blood samples were obtained at the same timepoint from both the coronary sinus and femoral vein from six patients with hypertrophic obstructive cardiomyopathy before undergoing alcohol septal ablation at Massachusetts General Hospital, as described previously[50].

**Aptamer-based plasma profiling**. Plasma samples from the five cohorts were randomly distributed across 96-well plates and underwent analysis with a multiplex assay based on protein capture with modified aptamers (slow-offrate modified aptamers, SOMAmers) detected and quantitated as relative fluorescence units (RFU) using oligonucleotide microarrays (Agilent Technologies, Santa Clara, CA)[10,12]. The aptamer platform (SOMAscan, SomaLogic Inc, Boulder, CA) allowed analysis of 1305 proteins, after exclusion of five aptamers as recommended by the supplier (alkaline phosphatase [bone], C1s, desmoglein-2, DR3, reticulon-4), representing a broad range of pathways and protein classes of which 47% are secreted proteins, 28% extracellular domains, 25% intracellular proteins. The alcohol septal ablation and BioVU cohorts were assayed using an earlier version of the assay containing aptamers for 1129 proteins, of which 1061 were identical with the more recent version[51]. The assay involves a multistep capture, release and recapture enrichment process, as described previously[12,50], and was performed semi-automatically using a liquid handling robot (Freedom EVO, Tecan Diagnostics, Switzerland) according to the manufacturers' detailed protocol. Each sample was diluted into three different concentrations (40%, 1%, 0.05%) to allow assessment of proteins at low, medium and high abundance in plasma, respectively, with a combined dynamic range of eight orders of magnitude. Each sample was further normalized based on (i) 12 hybridization control sequences on each microarray to correct for systematic variability in hybridization, and (ii) median signal based on all features for each dilution to correct for variability across plates. Acceptance criteria for individual samples relative to the median signal was between 0.4 and 2.5, based on historical runs. Each SOMAmer was further calibrated against five human pooled plasma samples on each 96-well plate. For plate calibration acceptance, 95% of SOMAmers must be less than 0.4 from the median within the plate. High specificity of selected SOMAmers and high reproducibility of RFU estimates has been described previously[11,50].

**Statistical analysis of protein data**. A total of 912 (70%) of the 1305 proteins showed a skewness coefficient based on moment around the mean statistic >2 and marked right-skew was confirmed for many upon histogram inspection, where all protein concentrations were natural logarithm transformed for uniformity. Protein levels were also scaled to a standard deviation of 1 for ease of interpretation.

The association of individual proteins with incident heart failure was tested in Cox proportional hazards regression models (583 population-representative controls, 185 cases) adjusting for age and sex, with Prentice weights to account for the case-cohort design[52], and robust variance estimators[53,54]. p values below a Bonferroni-adjusted threshold of 0.05, accounting for 1305 individual tests, were considered statistically significant ($p < 3.8 \times 10^{-5}$). The independence of proteins from conventional risk factors was tested by additional adjustment for established risk factors including body mass index, blood pressure (systolic, diastolic, use of antihypertensive therapy), cholesterol concentration (low density and high density), current smoking, history of diabetes and coronary heart disease, as described previously[15]. In a sensitivity analysis of nonischemic HF, follow-up was further censored at first myocardial infarction leaving 142 HF cases. Independence of renal function, sample plate and storage time was confirmed in individual models. Model assumptions of normality were confirmed for the logarithmically transformed proteins associated with HF by inspection of histograms, and the proportionality of hazards assumption was confirmed using Schoenfeld's global test. The pairwise correlation of proteins was examined using Pearson's correlation coefficients.

The association of proteins with manifest HF were tested by comparison of this cohort (84 cases) with the general MDCS cohort (583 population-representative controls), using unconditional logistic regression with bootstrap resampling for variance estimation, adjusted for age and sex. A secondary analysis also adjusted for body mass index, estimated glomerular filtration rate, history of diabetes and current smoking. To explore the external validity of our results, correlation of odds ratios per standard deviation of log-transformed RFUs and directional consistency was examined with results from the same logistic regression analysis in the small BioVU cohort (48 cases, 47 controls). Proteins associated with incident or manifest HF were compared in serial measures of 30 heart transplant recipients using the Wilcoxon signed-rank test. Protein measurements in coronary sinus and peripheral vein plasma from the same six patients were also compared descriptively for cardiac enrichment, for proteins associated with incident HF. Analyses were conducted using STATA version 12 (StataCorp, College Station, Texas, USA).

To test for common pathways enriched in the manifest HF association results, which may not be picked up through subjective manual annotation of findings, gene-set enrichment analysis[55] was conducted by testing proteins associated with HF for enrichment of hallmark gene sets annotated in the Molecular Signatures Database using hypergeometric tests as implemented in the GSEA software (http://software.broadinstitute.org/gsea/msigdb/index.jsp). FDR q values were computed according to Benjamini and Hochberg[56]. The hallmark gene set constitutes a computationally and manually curated version of the major functional annotation databases such as the Gene Ontology project, with reduced redundancy and heterogeneity to facilitate reproducibility of results as described previously[57].

**Spatial transcriptomic analysis of human hearts**. Tissue samples for spatial transcriptomic analysis were obtained from two explanted hearts from patients with ischemic cardiomyopathy and hypertrophic cardiomyopathy undergoing heart transplantation. Samples were directly fresh frozen and cryosectioned. Tissue sections were placed on Spatial Transcriptomics Library Preparation slides (Spatial Transcriptomics, Stockholm, Sweden) containing an array of reverse transcription primers with unique positional barcodes as described previously[24]. Sections were fixed, stained with Hematoxylin/Eosin and imaged before tissue permeabilization, RNA hybridization and cDNA synthesis. cDNA libraries were sequenced on an Illumina NextSeq 500 instrument and paired-end reads of 125 basepairs were aligned to the human genome using the public ST pipeline 1.6.2. A total of 50 million reads were sequenced. Alignment of the histological brightfield image with spots corresponding to each positional barcode was performed using the ST Spot Detector Software. Heterogeneity across spots and between patients was explored for proteins with evidence of cardiac origin, and colocalization was tested with other cell-type-specific transcripts.

**Genome-wide association studies**. Genotypes were available in 1421 subjects from the MDCS with SOMAscan, based on genome-wide genotyping using the Illumina Omni Express Exome BeadChip kit as described previously[26,38]. These subjects represented the 583 random cohort subset and 185 heart failure cases described here, and additional case subsets with cardiometabolic disease not contributing to the current study. Genotypes were called using Illumina GenomeStudio, and imputation was performed to the 1000 Genomes phase 1 version 3 (August 2012) reference panel using IMPUTE v2 for SNPs passing the following criteria: call rate ≥ 95%, pHWE ≥ $1 \times 10^{-6}$, and minor allele frequency ≥0.01. Genome-wide association analyses were conducted using general linear models with an additive inheritance model as implemented in the R package GWAF v2.2[58] (https://cran.r-project.org/web/packages/GWAF/index.html), adjusting for age and sex. Results meeting a $p$ value threshold of $p < 3 \times 10^{-9}$ were considered genome-wide significant, corresponding to Bonferroni correction for one million independent genetic variants and 16 tested proteins. In a second stage, a meta-analysis was conducted with a previously published study of 759 subjects from the Framingham Heart Study[26] for the 8 proteins where $cis$-pQTLs of genome-wide significance were not identified in the MDCS study (C5 [484 SNPs], CXCL13 [310 SNPs], DUSP3 [282 SNPs], gelsolin [239 SNPs], Nt-proBNP [484 SNPs], PRKACA [73 SNPs], tPA [188 SNPs], and UNC5D [834 SNPs]), MAF > 0.05 and imputation quality > 0.3 within ±1 Mb of the transcriptional start site. $p$ values below $2 \times 10^{-5}$ were considered statistically significant based on Bonferroni correction for 2894 tested SNPs.

**Reporting summary**. Further information on research design is available in the Nature Research Reporting Summary linked to this article.

## Data availability

All the relevant data supporting the findings of this study are available within this article, in the supplementary material, the source data file, or relevant repositories. Protein data are available within the source data file for all cohorts except the Framingham Heart Study, for which it has been deposited in the National Center for Biotechnology Information (NCBI) dbGaP repository: https://www.ncbi.nlm.nih.gov/gap/ (accession code pht006013.v2.p11). Genetic data for the Framingham Heart Study are also available through dbGaP (accession code phs000342.v18.p11). For the Swedish cohorts, Swedish and European legislation impose restrictions on public availability of datasets containing pseudonymized information. However, the full datasets including genome-wide data and phenotypes can be accessed through an institutional repository at Lund University with pertinent permissions (https://www.malmo-kohorter.lu.se/english). Spatial transcriptomic data have been deposited in the NCBI Gene Expression Omnibus database (GEO; accession code GSE135805).

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

## Acknowledgements

We acknowledge the contributions of the Region Skåne Biobank (Lund, Sweden) for storage and retrieval of the LCPR and MDCS samples, and of Science for Life Laboratory (Stockholm, Sweden) for RNA-sequencing of heart samples. This work was supported by grants (to J.G.S.) from the European Research Council (ERC-STG-2015-679242), the Swedish Research Council (2017-02554), the Swedish Heart-Lung Foundation (2016-0134 and 2016-0315), the Crafoord Foundation, Skåne University Hospital, the Scania county, governmental funding of clinical research within the Swedish National Health Service, a generous donation from the Knut and Alice Wallenberg foundation to the Wallenberg Center for Molecular Medicine in Lund, and funding from the Swedish Research Council (Linnaeus grant Dnr 349-2006-237, Strategic Research Area Exodiab Dnr 2009-1039) and Swedish Foundation for Strategic Research (Dnr IRC15-0067) to the Lund University Diabetes Center. This work was also supported by grants from NIH (R01HL132320 and R01HL133870 to T.J.W. and R.E.G., and K award 5K01GM103817 to D.N.), and the LaDue Foundation (to M.B.). Open access funding provided by Lund University.

## Author contributions

J.G.S. designed and supervised the study, and drafted the manuscript. J.B., R.N., Q.S.W., Q.Y., and J.G.S performed bioinformatic and statistical analyses. A.E., M.L.S., C.L., J.L., G.R., G.E., and J.G.S. collected and characterized phenotype data. M.B., Q.S.W., L.F., S.S., D.S., D.N., T.J.W, and R.E.G conducted the coronary sinus and validation studies. O.G. and S.C. performed the spatial transcriptomic analysis. A.E., J.B., M.L.S., O.G., R.N., M.B., Q.S.W., S.C., C.L., L.F., S.S., D.S., J.L., G.R., D.N., G.E., Q.Y., T.J.W., R.E.G., and J.G.S. reviewed the manuscript.

## Competing interests

The authors declare no competing interests.
