## [Peer Review File · Nature Communications]

Reviewers' comments:

Reviewer #2 (Remarks to the Author):

Egerstedt et al describe an aptamer-based proteomic screening effort in a series of very interesting plasma sample collections including 185 prospective heart failure (HF) incidence cases in a population-based cohort of 583 cases, a collection of 84 advanced HFs, 30 samples derived before/after heart transplantation, and 6 patients undergoing alcohol ablation for hypertrophic obstructive cardiomyopathy. A series of mostly binary comparisons was conducted to study the dynamics of 1,300 proteins in these different disease states. A major problem with a subset of these analyses is that blood samples were compared in some instances from very different sample collections and therefore the claim that 421 proteins among the 1,305 tested proteins were altered in the advanced heart failure cases is most likely confounded by technical artifacts. Most other analysis seem to be technically sound and in particular the proteins listed in Tables 2 and 3 that are associated with incident heart failure and heart transplantation are very interesting. Identifying pathways out of short lists of for example 16 proteins with 2 hits is not statistically sound, and the authors should think about not only highlighting pathways in the abstract, but also some of the proteins by name that are discussed at length in the discussion section. The manuscript is well written and additional analyses such as the origin and location of protein expression and the GWAS part add further value.

In prior studies, about 2/3s of all protein effects detected with aptamers were reproducible either with antibody-based or mass spectrometry-based assays (see Sun et al, Nature 2018 and Billing et al, Journal of Proteomics, 2017). As it is unclear for one third of aptamer measurements if the true target was identified by this single-binder assay, the authors should pick in particular newly identified proteins listed in Tables 2 and 3 and validate some of these with antibody- or mass spectrometry-based assays.

Major points:

Page 5, line 23: To demonstrate the quality of proteome measurements it would be important to show how the aptamer N-terminal pro-BNP and Brain natriuretic peptide 32 measurements compared to the NT-proBNP or BNP assays used in clinical routine for the same samples. For example, the data for NT-proBNP and CRP described on page 9, lines 1-4, should be shown as scatter plots in a Supplementary Figure: "The correlations of aptamer-based abundance estimates with

those of conventional immunoassays were high for both Nt-proBNP and CRP (Pearson's $r=0.66$ and 0.52 , respectively). Of the MMPs, only MMP7, MMP12 and none of the TIMPs were associated with incident HF."

Page 7, line 17: The authors observed that 421 proteins among the 1,305 tested proteins were altered in their advanced heart failure cases, by applying a Bonferroni-adjusted significance threshold of 0.05. Observing a significant change of 1/3 of all measured proteins seems extreme and the authors should further evaluate if a more stringent multiple testing approach such as Benjamini-Hochberg in conjunction with cut-offs scaled to corr. P-values of already known HF-relevant proteins may reduce this long list of candidate proteins to a more useful list. Since these advanced heart failure associated proteins are derived of a comparison of two very different cohorts, the authors should be careful with claiming that 1/3 of all measured proteins are significantly changing. Many of those changes may be due to pre-analytical technical artifacts.

Page 27, lines 10-14: "The association of proteins with manifest HF were tested by comparison of this cohort with the general MDCS cohort, using unconditional logistic regression with bootstrap resampling for variance estimation, adjusted for age, sex and BMI."

A main part of the results described in this study depend on a comparison of blood samples collected in different cohorts. The authors should state in what year and with what protocol samples were collected for MDCS and the manifest HF cohort and explain what differences there were in terms of collection protocols and what pre-analytical variables are to be expected because of that. For all statistical analyses descriptions in the Materials and Methods section the number of cases should be described for each test.

Page 8, lines 16-17: "We next studied the association of plasma proteins with development of HF during follow-up in subjects without HF at baseline in the nested case-cohort sample from the MDC-CC." The authors need to provide more information on how samples were collected at disease free baseline and after a HF event during follow-up. How much time has passed between the two samples and how much time after the HF event to 2nd sample collection?

Figure 2: Why only show Nt-proBNP and TSP2 and not any of the other heart transplantation affected proteins in Table 3, for example with changes $> \pm 50\%$?

Figures 3 and 4: Could benefit from encoding effect sizes/p-values of Heart failure associations in color scale for each node.

Minor Points:

Most Supplementary Tables: Should not be in a txt file, but actual xlsx or .txt tables.

Figure 1: Contains very little information and should be a Supplementary Figure.

Page 17, lines 5-7: "Activity in many of these pathways declined after heart transplantation, indicating that protein perturbations are to a large extent induced by the failing heart rather than comorbid conditions." Looking at Table 3, out of 16 proteins only 4 go down after heart transplantation. Pathway enrichment analysis out these small protein set is not feasible and this sentence should be changed to in "some of these proteins declined...".

Pages 24 and 25: The blood samples were derived of different body locations, with no information directly provided for the MDC-CC study, median cubital vein venepuncture for the advanced HF patients, central venous catheter-based collection for the heart transplantation cases and coronary sinus/femoral vein collection for the alcohol ablation cohort. The authors should comment on the expected/observed differences in blood sampling from different locations.

Page 9, lines 11-15: "To test for enrichment of pathways with HF development, we applied gene-set enrichment analysis to identify a total of 4 gene sets that were enriched among the 16 proteins (false discovery rate [FDR] <0.05%) as shown in Supplementary Table 8."

Gene set enrichment in a list of 16 proteins with 2 matches is not convincing. This should be taken out. Pathway annotations can be mentioned without statistics.

Page 14: lines 15-17: "Finally, to further explore the specificity of aptamer reagents and implicated pathways for the 16 proteins associated with incident HF, we conducted genome-wide association studies (GWAS) of these proteins in 1,421 subjects from the MDCS." The authors should comment on why for GWAS pQTL studies aptamer-based measurements of 1,421 cases were used and for the main analysis the number of cases was reduced to 583 cases?

Reviewer #3 (Remarks to the Author):

The authors report results from a multifaceted and massive exploratory protein biomarker discovery program for heart failure, performed in multiple sample banks, using the Somalogic platform. They measure over 1300 proteins in multiple patient samples, including a medium sized case control analysis from the Malmo population based study (n=583), a smaller cohort (n=84) with advanced heart failure and a very small (n=30) but interesting group with measurements before and after heart transplant. Finally, they measure CS and peripheral protein levels in 6 patients with HCM.

The authors are to be congratulated for their enormous effort, and there are many findings of potential significance.

Strengths of the paper include the replication of known proteins associated with heart failure, which is important for internal and external validity of the other findings. The inclusion of GWAS data

However, the findings are largely descriptive. Moreover, the paper is very difficult to follow given the sheer amount of data provided. If the authors want the paper to be read and digested instead of used as reference material, they will need to streamline and simplify the presentation. As written, I can see the paper of clear value for investigators wanting to find out if "anyone has ever looked at protein X in heart failure" but for a general reader the key scientific insights are more challenging to decipher.

Comments

1. Although the authors title the paper, "A dynamic view of the plasma proteome in human heart failure" this is misleading, as only the heart transplant studies provide a dynamic picture. A stronger design would indeed have been to perform serial measurements in the same individuals over time. Here, however, the comparisons are mostly cross sectional comparisons that are often performed across different cohorts. The only group where serial measurement is done is the transplant patients. The title should be changed and this should be modified throughout.

2. A major limitation is the absence of adjustment for renal function. Given important differences between groups in renal function, and the known effect of renal clearance on circulating protein levels, some/many of the findings may be related to renal clearance rather than true differences in protein secretion or release. This must be addressed in revision.

3. I would remove the analyses from the HCM samples. First, this is a different disease state. Second, the number of samples is so small it is difficult to have confidence. Finally, it is yet another experiment in an over crowded paper and I think adds little. Finally, the methods are unclear as it suggests that the CS measurements were performed in the transplant group too.
4. I'm worried that the use of extremely strict p values (bonferroni corrected) for discovery may lead to omission of relevant markers and pathways from consideration as important. In my view a biomarker that replicates across the relevant studies, even if the p value did not meet initial bonferroni criteria, would be of interest.
5. Please reorder the supplement (or consider 2 supplements) so the the reference material is at the end and the important material first
6. Please explain the rationale for the gene set enrichment analyses more clearly, as the relevance of these experiments may not be obvious to all readers.
7. Too little attention is given to the large number of proteins with concentrations that are LOWER in HF and decrease after transplant. Please enhance discussion of the relevance of these pathways as loss of protective pathways may be as important as emergence of detrimental ones.
8. Clarify if the Malmo studies are case control or case-cohort.

Reviewer #4 (Remarks to the Author):

Egerstedt and colleagues test the association of >1000 plasma proteins with incident heart failure. They report that 16 proteins associate with heart failure.

Strengths:

1. Prospective cohort design
2. Large number of plasma protein measurements

Major Limitations:

1. For the prediction analyses, there is no replication in a cohort independent of the Malmo study.
2. No clear, new, robust biologic insight.

Minor Limitations:

1. The analyses presented in cohorts other than MDC (i.e., advanced HF, post-transplant, and septal ablation cases) do not seem robust and these sections read as quite anecdotal.
2. The GWAS for the 16 plasma proteins seems misplaced. With these data, what insights do you establish into heart failure?
3. Are Figures 3 and 4 based on data from the study or out of textbooks?

Response to reviewers

Reviewer #2 (Remarks to the Author):

Egerstedt et al describe an aptamer-based proteomic screening effort in a series of very interesting plasma sample collections including 185 prospective heart failure (HF) incidence cases in a population-based cohort of 583 cases, a collection of 84 advanced HFs, 30 samples derived before/after heart transplantation, and 6 patients undergoing alcohol ablation for hypertrophic obstructive cardiomyopathy. A series of mostly binary comparisons was conducted to study the dynamics of 1,300 proteins in these different disease states. A major problem with a subset of these analyses is that blood samples were compared in some instances from very different sample collections and therefore the claim that 421 proteins among the 1,305 tested proteins were altered in the advanced heart failure cases is most likely confounded by technical artifacts. Most other analysis seem to be technically sound and in particular the proteins listed in Tables 2 and 3 that are associated with incident heart failure and heart transplantation are very interesting. Identifying pathways out of short lists of for example 16 proteins with 2 hits is not statistically sound, and the authors should think about not only highlighting pathways in the abstract, but also some of the proteins by name that are discussed at length in the discussion section. The manuscript is well written and additional analyses such as the origin and location of protein expression and the GWAS part add further value.

In prior studies, about 2/3s of all protein effects detected with aptamers were reproducible either with antibody-based or mass spectrometry-based assays (see Sun et al, Nature 2018 and Billing et al, Journal of Proteomics, 2017). As it is unclear for one third of aptamer measurements if the true target was identified by this single-binder assay, the authors should pick in particular newly identified proteins listed in Tables 2 and 3 and validate some of these with antibody- or mass spectrometry-based assays.

Author response: We appreciate the thoughtful comments. We agree that target binding is an important issue to address. To this end, we have now included additional information on correlations with dual-binder immunoassays (confirming correlation for all nine of the 16 incident HF-associated proteins that were available on both platforms), and expanded genetic analyses (identifying associations in cis for 13 of the 16 proteins), in total presenting evidence of target binding for 14 of the 16 proteins (all except PRKACA and DUSP3) in **Table 4** and **Supplementary Figure 3**. We were unable to find any data confirming or refuting target binding of DUSP3 or PRKACA, as described under the section “Genome-wide association studies of plasma proteins”. With regard to the analysis of advanced heart failure, we have updated the manuscript with detailed discussion of pre-analytical factors that may influence the results (under Methods and Discussion), and have conducted additional analyses with further adjustments (**Supplementary Table 8**). We have also been able to include data from an external cohort, with heart failure cases and controls from the BioVU biorepository, which corroborate our results (described under “Plasma proteins in advanced heart failure”). We agree however that some extent of unmeasured confounding, as always, cannot be ruled out and have therefore also reduced emphasis on some of the conclusions related to the advanced heart failure analysis in the manuscript (abstract and discussion). Finally, the pathway analysis for incident HF has been removed from the manuscript, focusing instead in the discussion on what is known about the individual proteins identified (naming the most important ones in the abstract).

Major points:

Page 5, line 23: To demonstrate the quality of proteome measurements it would be important to show how the aptamer N-terminal pro-BNP and Brain natriuretic peptide 32 measurements compared to the NT-proBNP or BNP assays used in clinical routine for the same samples. For example, the data for NT-proBNP and CRP described on page 9, lines 1-4, should be shown as scatter plots in a Supplementary Figure: “The correlations of aptamer-based abundance estimates with those of conventional immunoassays were high for both Nt-proBNP and CRP (Pearson’s $r=0.66$ and 0.52 , respectively). Of the MMPs, only MMP7, MMP12 and none of the TIMPs were associated with incident HF.”

Author response: We have included scatter plots for proBNP and CRP, where clinical immunoassays were available, in **Supplementary Figure 3**. An immunoassay for BNP was not available to us. We also included scatter plots in the same figure for the 7 other proteins where we obtained information from Somascan proteins and a dual-binder immunoassay (proximity extension assay, Olink Proteomics).

Page 7, line 17: The authors observed that 421 proteins among the 1,305 tested proteins were altered in their advanced heart failure cases, by applying a Bonferroni-adjusted significance threshold of 0.05. Observing a significant change of 1/3 of all measured proteins seems extreme and the authors should further evaluate if a more stringent multiple testing approach such as Benjamini-Hochberg in conjunction with cut-offs scaled to corr. P-values of already known HF-relevant proteins may reduce this long list of candidate proteins to a more useful list. Since these advanced heart failure associated proteins are derived of a comparison of two very different cohorts, the authors should be careful with claiming that 1/3 of all measured proteins are significantly changing. Many of those changes may be due to pre-analytical technical artifacts.

Author response: We agree that the finding of widespread perturbation of the plasma proteome in advanced heart failure merits further attention. As described above, we have updated the manuscript with detailed discussion on pre-analytical factors that differ between cohorts and may influence the results, and have conducted additional analyses with further adjustments. We have also included data from an independent cohort, cases and controls from the BioVU biorepository. These data, although of limited sample size (48 cases and 47 controls) and consequently with low power to replicate associations for individual proteins, displayed consistent effect direction and correlation with effects from the MDCS study (described under “Plasma proteins in advanced heart failure”). Importantly, we believe that these correlations of our study and bioVU argues against a technical artifact, but we agree that some extent of unmeasured confounding cannot be ruled out and have therefore also reduced emphasis on some of the conclusions related to the advanced heart failure analysis in the manuscript (abstract and discussion). With regards to re-scaling of p-values, we have elected not to re-scale based on the strong well-established biomarkers such as proBNP which would result in an arbitrary loss of association with biomarkers of more modest effect, but rather conducted the analyses described above and provide the complete list of results in **supplementary table 8**. Finally, it is worth emphasising that given the nature of this severe disease, with

hemodynamic consequences influencing most organ systems, we do not view this finding as biologically implausible.

Page 27, lines 10-14: “The association of proteins with manifest HF were tested by comparison of this cohort with the general MDCS cohort, using unconditional logistic regression with bootstrap resampling for variance estimation, adjusted for age, sex and BMI.”

A main part of the results described in this study depend on a comparison of blood samples collected in different cohorts. The authors should state in what year and with what protocol samples were collected for MDCS and the manifest HF cohort and explain what differences there were in terms of collection protocols and what pre-analytical variables are to be expected because of that. For all statistical analyses descriptions in the Materials and Methods section the number of cases should be described for each test.

Author response: We have clarified under Methods the sampling timepoint and protocol for each sample. The sampling protocols and biobank management was similar across cohort, but two major preanalytical factors differed between the HF cases and population cohort: the fasting conditions and storage times. Specifically, all population-based samples were obtained after an overnight fast while fasting conditions were unclear for HF samples. However, the impact of fasting on the 1305 somascan proteins has been described previously to be limited (Kim CH et al, Sci Rep 2018;8:8382), with a median difference of only 1% and no significant difference detected for any protein in that study. Regarding storage time, they were substantially longer in the control group (population cohort from the 90s) than from the case group (2010s). However, the largely consistent effect directionality and effect correlation with the independent BioVU cohort argue against a major impact of preanalytical factors. Unmeasured confounding on the other hand cannot be ruled out, as always. We have included a discussion of preanalytical factors and confounding in the penultimate paragraph of the Discussion, and further highlighted the different cohorts in a new **Figure 1**. We have also included the number of cases and controls included for each test under “Statistical analysis of protein data” under Methods.

Page 8, lines 16-17: “We next studied the association of plasma proteins with development of HF during follow-up in subjects without HF at baseline in the nested case-cohort sample from the MDC-CC.” The authors need to provide more information on how samples were collected at disease free baseline and after a HF event during follow-up. How much time has passed between the two samples and how much time after the HF event to 2nd sample collection?

Author response: We have clarified that samples from patients with manifest, advanced HF derived from an independent, clinical cohort under Methods and Results and in a new figure (Figure 1). We have also added additional detail on cohort descriptions and sampling conditions for the two cohorts.

Figure 2: Why only show Nt-proBNP and TSP2 and not any of the other heart transplantation affected proteins in Table 3, for example with changes >+/- 50%?

Author response: We have now updated Figure 2 to include the 6 proteins from Table 3 with change >+/-50%.

Figures 3 and 4: Could benefit from encoding effect sizes/p-values of Heart failure associations in color scale for each node.

Author response: We have updated the figures with color coding and agree this made them easier to follow. Red color now indicates significantly higher concentrations in subjects with incident HF, green indicates lower, yellow indicates no significant difference, and white indicates that the protein was not available on the assay.

Minor Points:

Most Supplementary Tables: Should not be in a txt file, but actual xlsx or .txt tables.

Author response: We have moved the Supplementary Tables to an xlsx file and agree this makes it much easier to read them.

Figure 1: Contains very little information and should be a Supplementary Figure.

Author response: We have moved Figure 1 to the supplement (now Supplementary Figure 2). We have replaced it with a Figure that illustrates the natural history of heart failure and the differences between the cohorts studied here.

Page 17, lines 5-7: "Activity in many of these pathways declined after heart transplantation, indicating that protein perturbations are to a large extent induced by the failing heart rather than comorbid conditions." Looking at Table 3, out of 16 proteins only 4 go down after heart transplantation. Pathway enrichment analysis out these small protein set is not feasible and this sentence should be changed to in "some of these proteins declined...".

Author response: We have rephrased the sentence.

Pages 24 and 25: The blood samples were derived of different body locations, with no information directly provided for the MDC-CC study, median cubital vein venepuncture for the advanced HF patients, central venous catheter-based collection for the heart transplantation cases and coronary sinus/femoral vein collection for the alcohol ablation cohort. The authors should comment on the expected/observed differences in blood sampling from different locations.

Author response: We have clarified under Methods that blood samples were similarly obtained through median cubital vein venipuncture into EDTA tubes in the MDC-CC and advanced HF. In contrast, heart transplant recipients and the alcohol ablation cohort had sampling from other locations but were only used for within-cohort comparisons. We have clarified this in the penultimate paragraph of the Discussion, and that substantial differences in circulating levels can be expected for some proteins between vascular territories and between venous and arterial samples, as proteases and clearance receptors rapidly clear some proteins from circulation. For the figure illustrating protein distributions across the cohorts studied here (Figure 2), we note directional consistency between the advanced HF cohort and heart transplant recipients, but now also recognize in the limitations paragraph of the discussion that the impact of differences between vascular beds remains unclear.

Page 9, lines 11-15: “To test for enrichment of pathways with HF development, we applied gene-set enrichment analysis to identify a total of 4 gene sets that were enriched among the 16 proteins (false discovery rate [FDR] <0.05%) as shown in Supplementary Table 8.”

Gene set enrichment in a list of 16 proteins with 2 matches is not convincing. This should be taken out. Pathway annotations can be mentioned without statistics.

Author response: We have removed the analysis (supplementary table) and instead discuss annotations for the identified individual proteins, as suggested. We have kept the pathway analysis for advanced HF, where we had 421 proteins.

Page 14: lines 15-17: “Finally, to further explore the specificity of aptamer reagents and implicated pathways for the 16 proteins associated with incident HF, we conducted genome-wide association studies (GWAS) of these proteins in 1,421 subjects from the MDCS.” The authors should comment on why for GWAS pQTL studies aptamer-based measurements of 1,421 cases were used and for the main analysis the number of cases was reduced to 583 cases?

Author response: We thank the reviewer for pointing to this numeric inconsistency. We have now clarified in the manuscript that GWAS was available in the entire MDCS cohort, but somascan was only available in a case-cohort set of 1421 of these subjects representing a random cohort subset of 583 subjects, 185 incident heart failure cases, and a number of additional case subsets for various diseases not contributing to the current study.

Reviewer #3 (Remarks to the Author):

The authors report results from a multifaceted and massive exploratory protein biomarker discovery program for heart failure, performed in multiple sample banks, using the Somalogic platform. They measure over 1300 proteins in multiple patient samples, including a medium sized case control analysis from the Malmo population based study (n=583), a smaller cohort (n=84) with advanced heart failure and a very small (n=30) but interesting group with measurements before and after heart transplant. Finally, they measure CS and peripheral protein levels in 6 patients with HCM.

The authors are to be congratulated for their enormous effort, and there are many findings of potential significance. Strengths of the paper include the replication of known proteins associated with heart failure, which is important for internal and external validity of the other findings. The inclusion of GWAS data

However, the findings are largely descriptive. Moreover, the paper is very difficult to follow given the sheer amount of data provided. If the authors want the paper to be read and digested instead of used as reference material, they will need to streamline and simplify the presentation. As written, I can see the paper of clear value for investigators wanting to find out if "anyone has ever looked at protein X in heart failure" but for a general reader the key scientific insights are more challenging to decipher.

Author response: We thank the reviewer for interest in our manuscript and thoughtful comments. We believe that it is a strength rather than a limitation that our analysis points to a full and complex landscape of proteomic associations reflecting diverse pathophysiologies, rather than one or two proteins, but we have attempted to streamline the presentation and move some content to the supplement. Furthermore, we have revised the supplement into an excel document for improved overview, as suggested by reviewer 2.

Comments

1. Although the authors title the paper, "A dynamic view of the plasma proteome in human heart failure" this is misleading, as only the heart transplant studies provide a dynamic picture. A stronger design would indeed have been to perform serial measurements in the same individuals over time. Here, however, the comparisons are mostly cross sectional comparisons that are often performed across different cohorts. The only group where serial measurement is done is the transplant patients. The title should be changed and this should be modified throughout.

Author response: We agree that serial samples were only obtained from the transplant patients and that the term "dynamic" may give the wrong impression of the study design. We have changed the title and wording throughout the manuscript to: "Profiling of the plasma proteome across different stages of human heart failure". To illustrate the study design, we have replaced the previous Figure 1 (p-value histogram, moved to supplementary figure 2) with a schematic overview of the natural history of heart failure and indicated the timepoints of sampling and sample sizes for the cohorts in the study, which correspond to the different cohorts across which individual biomarker distributions are depicted in Figure 2.

2. A major limitation is the absence of adjustment for renal function. Given important differences between groups in renal function, and the known effect of renal clearance on circulating protein levels, some/many of the findings may be related to renal clearance rather than true differences in protein secretion or release. This must be addressed in revision.

Author response: We agree that adjustment for renal function is warranted. We elected not to include this adjustment in the primary analysis, as the bidirectional relationship with cardiac function (i.e. heart failure begets renal failure and renal failure begets heart failure) make interpretation of such adjustment difficult. We have now conducted additional secondary analyses, adjusting for renal function (see updated Table 2 for incident heart failure and updated Supplementary Table 8 for manifest heart failure). In summary, this additional adjustment resulted in only minimal changes in effect estimates for both the 16 proteins associated with incident heart failure and for the 421 associated with manifest HF.

3. I would remove the analyses from the HCM samples. First, this is a different disease state. Second, the number of samples is so small it is difficult to have confidence. Finally, it is yet another experiment in an over crowded paper and I think adds little. Finally, the methods are unclear as it suggests that the CS measurements were performed in the transplant group too.

Author response: We have reduced emphasis on the HCM analyses, and now only include it in the supplement and briefly in the examination of tissue origin of proteins. Importantly, the findings of that analysis is consistent with findings from GTEx, Human Protein Atlas, Cardiac mass spectrometry and the Tabula Muris (which has now been added) resources in implicating a cardiac origin for Nt-proBNP, thrombospondin-2, IL-18 receptor, and gelsolin, and provides interesting suggestive evidence of higher activated C5 in coronary blood. We clarify under discussion (penultimate paragraph) that this is a distinct pathophysiological condition.

4. I'm worried that the use of extremely strict p values (bonferroni corrected) for discovery may lead to omission of relevant markers and pathways from consideration as important. In my view a biomarker that replicates across the relevant studies, even if the p value did not meet initial bonferroni criteria, would be of interest.

Author response: We agree with the reviewer. We have therefore made the complete results lists available in the supplement (Supplementary tables 5 and 8).

5. Please reorder the supplement (or consider 2 supplements) so the the reference material is at the end and the important material first

Author response: We have now created two supplements: we moved all supplementary tables to an excel file, as also suggested by reviewer 2. The reference materials are thus in a separate sheet from more important tables. The Supplementary Figures are in a separate word file. We agree this makes the supplementary content much easier to read.

6. Please explain the rationale for the gene set enrichment analyses more clearly, as the relevance of these experiments may not be obvious to all readers.

Author response: We have clarified (under Methods and Results) that the gene set enrichment is a data-driven approach to identify common pathways from the large number of proteins associated with manifest heart failure. In this context, manual annotation of results is infeasible why pathway analysis may identify pathophysiological mechanisms that may otherwise be missed. We have removed this analysis for incident HF as suggested by reviewer 2, where we only had 16 proteins, but include the analysis for advanced HF where we have 421 proteins.

7. Too little attention is given to the large number of proteins with concentrations that are LOWER in HF and decrease after transplant. Please enhance discussion of the relevance of these pathways as loss of protective pathways may be as important as emergence of detrimental ones.

Author response: We have clarified under Results and Discussion that we are uncertain as to the importance of most of the identified proteins with inverse associations. Our data suggests that most of the pathways that are inversely associated with HF are likely not protective pathways but rather unspecific with wide tissue expression and intracellular or membrane-bound (and thus should not be present in circulation). We have however clarified that important protective pathways may be present amongst those that may reflect unspecific processes such as tissue hypoperfusion.

8. Clarify if the Malmo studies are case control or case-cohort.

Author response: We have clarified the differences between included cohorts in a new Figure (Figure 1) as well as under Results and Methods. We describe that the prospective cohort from Malmö Diet and Cancer study (MDCS) represents a nested case-cohort design, while the advanced HF and transplant cases come from a clinical collection. We utilize the random population-based cohort from the MDCS as controls for the advanced HF cases, while the transplant cases are their own controls through serial sampling.

Reviewer #4 (Remarks to the Author):

Egerstedt and colleagues test the association of >1000 plasma proteins with incident heart failure. They report that 16 proteins associate with heart failure.

Strengths:

1. Prospective cohort design
2. Large number of plasma protein measurements

Major Limitations:

1. For the prediction analyses, there is no replication in a cohort independent of the Malmo study.

Author response: We thank the reviewer for interest in our manuscript and thoughtful comments. We agree that replication would have been desirable, but have only been able to obtain a small validation cohort for the manifest heart failure analysis, with correlation of effect estimates.

2. No clear, new, robust biologic insight.

Author response: We believe that several important findings emerged from our study, but that detailed mechanistic explorations of the identified proteins will need to be conducted as follow-up to this work. Two particularly interesting findings were the robust association with thrombospondin-2, which also came out of an early gene expression screen (Schroen B et al, Circ Res 2004) but that has received limited attention, and with activation of the complement cascade before HF onset which was recently implicated in myocardial repair mechanisms (Natarajan N et al, Circulation 2018). Additional interesting findings include inflammatory markers which have received limited attention in HF, including CXCL13 and IL18R. We have now named these markers in the abstract, to further highlight them. Finally, beyond the limitations confined by strict significance thresholds in a discovery effort, our results provide an atlas of changes in plasma proteins across different stages of heart failure that constitutes a resource for future investigations, as the full results are made available. We note many interesting candidate proteins at sub-threshold p-values and provide the full list of results with the manuscript.

Minor Limitations:

1. The analyses presented in cohorts other than MDC (i.e., advanced HF, post-transplant, and septal ablation cases) do not seem robust and these sections read as quite anecdotal.

Author response: We agree that the post-transplant cohort is of limited sample size but believe that it may be informative with regards to large effects. We have further explored the external validity of the advanced HF findings in an independent cohort and observe mostly directional consistency of effects and correlation of effects, and have conducted additional statistical adjustments as also suggested by the other reviewers. The septal ablation cohort has been toned down, but we note that the findings of that analysis is consistent with findings from GTEX, Human Protein Atlas, Cardiac mass

spectrometry and the Tabula Muris (which has now been added) resources in implicating a cardiac origin for Nt-proBNP, thrombospondin-2, IL-18 receptor, and gelsolin, and provides interesting suggestive evidence of higher activated C5 in coronary blood (Results and Discussion).

2. The GWAS for the 16 plasma proteins seems misplaced. With these data, what insights do you establish into heart failure?

Author response: We believe that this analysis contributes important information on the target binding of the aptamer reagents, in identifying cis-regulatory variants for 13 of 16 proteins associated with incident HF, as now further clarified under Discussion. In addition, the observation that GWAS of C5a finds CFH, which is central to C5 activation in the alternative pathway, provides further support for a role of that pathway for the measured concentration of C5a. We have now also supplemented the manuscript with target binding validation through correlations with immunoassays for 9 of the 16 aptamers. We have rephrased the abstract to focus less on the GWAS, so as not to confuse the reader regarding the role of these analyses.

3. Are Figures 3 and 4 based on data from the study or out of textbooks?

Author response: We have clarified that the pathways onto which the association results are mapped are considered well-established and were based on published review articles. We have expanded the figures to include color-coded effects from our study, as suggested by reviewer 2.

REVIEWERS' COMMENTS:

Reviewer #2 (Remarks to the Author):

The authors have addressed all points of concern properly.

Reviewer #3 (Remarks to the Author):

The authors have done an excellent job addressing my comments from the initial submission. The aggregate work here represents a massive undertaking, and while exploratory and descriptive, provides multiple new potential insights into heart failure pathophysiology.

In their final uploaded files, I would suggest the authors increase the font size for axis labels, particularly in the supplements, as they are hard to read.

James de Lemos

Reviewer #4 (Remarks to the Author):

The main limitation I outlined was lack of replication for the prediction analyses in the Malmo study. For omics studies, replication has emerged as an absolute (and basic) requirement.

Why not able to provide replication in an independent cohort study? Several NHLBI prospective cohorts (eg Framingham, Jackson Heart Study, ARIC) have these same Somalogic measurements and have incident heart failure curated.

Response to reviewers

Reviewer #2 (Remarks to the Author):

The authors have addressed all points of concern properly.

Author response: We again wish to thank the reviewer for interest in our manuscript and thoughtful comments.

Reviewer #3 (Remarks to the Author):

The authors have done an excellent job addressing my comments from the initial submission. The aggregate work here represents a massive undertaking, and while exploratory and descriptive, provides multiple new potential insights into heart failure pathophysiology.

In their final uploaded files, I would suggest the authors increase the font size for axis labels, particularly in the supplements, as they are hard to read.

James de Lemos

Author response: We again wish to thank Dr De Lemos for interest in our manuscript and thoughtful comments. We have increased figure sizes for the supplemental figures that we agree were difficult to read.

Reviewer #4 (Remarks to the Author):

The main limitation I outlined was lack of replication for the prediction analyses in the Malmo study. For omics studies, replication has emerged as an absolute (and basic) requirement.

Why not able to provide replication in an independent cohort study? Several NHLBI prospective cohorts (eg Framingham, Jackson Heart Study, ARIC) have these same Somalogic measurements and have incident heart failure curated.

Author response: We agree that the lack of replication in an independent cohort represents a limitation to our work and have clarified this limitation in the discussion section of the manuscript. We have also made sure that no claims regarding utility of these proteins for risk prediction are made in the manuscript, in the absence of replication.